# Age-dependent electroencephalogram (EEG) patterns during sevoflurane general anesthesia in infants

Laura Cornelissen[1][*][†], Seong-Eun Kim[2][†], Patrick L Purdon[3][‡], Emery N Brown[2,3][‡], Charles B Berde[1][‡]

[1]Department of Anesthesiology, Perioperative and Pain Medicine, Boston Children's Hospital, Boston, United States; [2]Department of Brain and Cognitive Sciences, Massachusetts Institute of Technology, Cambridge, United States; [3]Department of Anesthesia, Critical Care, and Pain Medicine, Massachusetts General Hospital, Boston, United States

**Abstract** Electroencephalogram (EEG) approaches may provide important information about developmental changes in brain-state dynamics during general anesthesia. We used multi-electrode EEG, analyzed with multitaper spectral methods and video recording of body movement to characterize the spatio-temporal dynamics of brain activity in 36 infants 0–6 months old when awake, and during maintenance of and emergence from sevoflurane general anesthesia. During maintenance: (1) slow-delta oscillations were present in all ages; (2) theta and alpha oscillations emerged around 4 months; (3) unlike adults, all infants lacked frontal alpha predominance and coherence. Alpha power was greatest during maintenance, compared to awake and emergence in infants at 4–6 months. During emergence, theta and alpha power decreased with decreasing sevoflurane concentration in infants at 4–6 months. These EEG dynamic differences are likely due to developmental factors including regional differences in synaptogenesis, glucose metabolism, and myelination across the cortex. We demonstrate the need to apply age-adjusted analytic approaches to develop neurophysiologic-based strategies for pediatric anesthetic state monitoring.

*For correspondence: laura.cornelissen@childrens.harvard.edu

†These authors contributed equally to this work

‡These authors are second equal contributors

## Introduction

In the United States, 200,000 children a year receive general anesthesia during the first year of life (*Rabbitts et al., 2010*). Anesthetics have profound effects on all physiological systems. As a consequence, since the mid-1980s, anesthesia caregivers have been required to monitor blood pressure, heart rate, body temperature, and oxygen saturation along with anesthetic gas and oxygen delivery for all patients receiving general anesthesia. The states of general anesthesia and sedation are produced by the anesthetics acting in the brain and spinal cord (*Brown et al., 2010, 2011; Ching and Brown, 2014*). However, brain monitoring using electroencephalogram (EEG)-derived indices is used to only a limited extent in adults, rarely in children, and essentially not at all in infants. These EEG-derived indices, which been developed in adults can give inaccurate indications of anesthetic states in infants and younger children (*Davidson, 2007; Constant and Sabourdin, 2012*). The lack of principled strategies for monitoring the brains of infants and children receiving anesthesia care is especially troubling in view of growing concern about anesthetic toxicity to the developing brain (*McCann and Soriano, 2012; Jevtovic-Todorovic et al., 2013; Lin et al., 2014*).

A plausible inference to be drawn from the few available EEG studies of children under general anesthesia is that the inaccuracy of EEG-derived indices in pediatric practice is likely due to differences between children and adults in their brain responses to the anesthetics. A consensus has

**eLife digest** Every year about 200,000 infants in the United States are given general anesthesia during their first year of life. Though anesthesia is essential to control pain during surgery and other medical procedures on infants, it involves some risks. There are some controversial studies suggesting that repeated anesthetics early in life may impact how the brain develops, but other studies have been reassuring and found no such effects. To reduce the risks, doctors carefully monitor infants' blood pressure, heart rate, body temperature, and oxygen levels while they are receiving anesthesia.

Electroencephalograms (EEGs) have proven to be a useful tool for monitoring the brain activity of adults undergoing anesthesia, but studies have found EEG-based monitoring to be unreliable in infants under anesthesia. A more reliable method of monitoring the brains of infants during anesthesia is needed. Anesthesiologists nevertheless need to better understand how the infant's brain works under general anesthesia, and novel EEG techniques hold promise for monitoring brain well-being and for adjusting anesthetic dosing in infants of different ages.

Differences in the way infants' brains respond to anesthesia may explain why current EEG-based monitoring methods developed for adults don't work as well in infants as in adults. Now, Cornelissen, Kim et al. have used a new EEG-based approach to demonstrate that as infants' brains develop, their responses to anesthesia change. The experiments involved 36 infants aged up to six months old who were going through routine surgical procedures. The brain activity of the infants was recorded using EEG—via electrodes placed on their scalps—when they were awake, during anesthesia, and as they recovered afterwards. The infants were videoed at the same time. Comparing the video with the EEG recordings allowed the brain activity of the infants to be matched up with their state of consciousness.

Cornelissen, Kim et al. detected slow waves of brain activity across the entire scalp of infants who are under six months old and under anesthesia. Infants who are older than about four months old also display some faster brain waves, which decreased in power as the infants emerged from anesthesia. However, none of the infants has the same pattern seen in adults—where faster waves appear near the front of the brain. These findings may help scientists develop more reliable ways to monitor infants' brains during anesthesia.

not been achieved on what these differences are because the available pediatric investigations have studied a limited set of anesthetics and frequently used EEG montages with few electrodes; different analysis methods have been used in different studies; multi-electrode recordings available in children have not been analyzed in function of age; and dose-titration experiments commonly conducted in adults cannot for ethical reasons be conducted in children (*Davidson et al., 2008*; *Lo et al., 2009*; *Hayashi et al., 2012*; *McKeever et al., 2012*; *Sury et al., 2014*).

For adults, characterizing the anesthetic response of the brain under general anesthesia and sedation is an active research field in which EEG, intracranial recordings, functional magnetic resonance imaging, and positron emission tomography are being used to study spatio-temporal changes in brain activity for different anesthetics and for different doses of the same anesthetic (*Feshchenko et al., 2004*; *Boveroux et al., 2010*; *Breshears et al., 2010*; *Mhuircheartaigh et al., 2010*; *Cimenser et al., 2011*; *Martuzzi et al., 2011*; *Murphy et al., 2011*; *Boly et al., 2012*; *Lewis et al., 2012*; *Casali et al., 2013*; *Lee et al., 2013*; *Liu et al., 2013*; *Ní Mhuircheartaigh et al., 2013*; *Purdon et al., 2013*; *Akeju et al., 2014a*; *Vizuete et al., 2014*). For example, a significant EEG pattern observed in adults under propofol general anesthesia and anesthesia maintained by an ether-derived anesthetic is incoherent slow oscillations across the entire head and strongly coherent alpha oscillations across only the front of the head (*Purdon et al., 2013*). Modeling studies suggest that the alpha oscillations are thalamo-cortical in origin (*Vijayan et al., 2013*). In addition, it is now appreciated that different anesthetics have different EEG signatures that are readily visible in the unprocessed EEG and its spectrogram (*Purdon et al., 2013*; *Akeju et al., 2014b*, *2014c*). These EEG signatures can be related to the altered arousal states the anesthetics produce and to the mechanisms through which the drugs are believed to act at specific receptors in specific brain circuits.

These neurophysiological studies in adults have led to the idea of using the unprocessed EEG and its spectrogram as an alternative to using EEG-derived indices to monitor the brain states of adults receiving general anesthesia or sedation (*Brown et al., 2015*).

Use of the unprocessed EEG and its spectrogram to develop a neurophysiological-based paradigm tailored to pediatric practices for monitoring anesthetic state requires studying in children the spatio-temporal dynamics of brain responses to anesthetics. To begin to address this key knowledge gap, we used multi-electrode EEG recordings to study the spatial and temporal dynamics of brain activity in infants 0–6 months of age when awake, and during maintenance of and emergence from sevoflurane general anesthesia administered for routine surgical care.

## Results

Continuous multichannel EEG recordings were collected during the awake state, Maintenance Of a Surgical State of Anesthesia (MOSSA), and emergence from sevoflurane general anesthesia in 36 infants aged 0–6 months postnatal age (*Figure 1*). Infant demographics and clinical characteristics are given in *Table 1*. Details concerning study procedures and enrollment are given in the 'Materials and methods' and *Figure 2*.

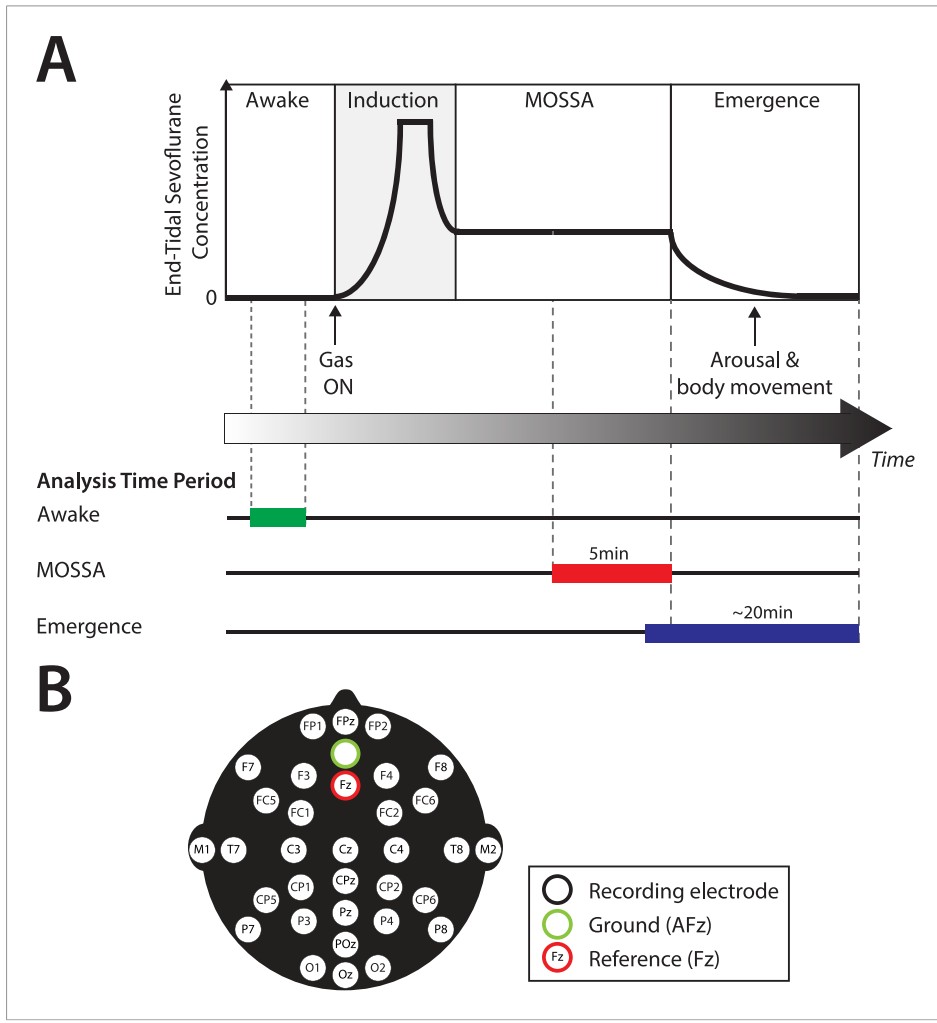

**Figure 1**. Experiment design. (**A**) Experiment timeline: schematic time-course of end-tidal sevoflurane concentration during the awake phase, induction, MOSSA (Maintenance of Surgical State of Anesthesia), and emergence phases of general anesthesia. Electroencephalogram (EEG) data from individual recording electrodes were analyzed post hoc for the awake brain state prior to anesthesia (shown in green), and two phases of general anesthesia (i) MOSSA (shown in red), and (ii) emergence from general anesthesia (shown in blue). (**B**) EEG montage used (modified international 10/20 electrode placement system).

**Table 1.** Infant demographics and clinical characteristics

| | All infants (N = 30) | 0–3 M (n = 11) | 4–6 M (n = 19) | Difference between medians in each age group | p-value |
|---|---|---|---|---|---|
| **Demographics** | | | | | |
| PMA at birth (weeks)* | 39.0 (CI: 37.6–39.0) | 37.0 (CI: 34.0–39.0) | 39.0 (CI: 39.0–39.0) | 2.0 | **0.02** |
| PNA at study (months)* | 5.5 (CI: 3.6–6.1) | 2.8 (CI: 0.5–3.5) | 6.1 (CI: 5.5–6.1) | 3.3 | – |
| Weight at study (kg)* | 6.8 (CI: 6.1–7.6) | 5.0 (CI: 4.1–6.3) | 7.6 (CI: 6.7–8.2) | – | – |
| Male [%, (n)]† | 83.3 (25) | 72.7 (8) | 89.5 (17) | – | 0.24 |
| **Procedure type** | | | | | |
| General surgery [%, (n)] | 53.3 (16) | 90.9 (10) | 31.6 (6) | – | – |
| Urological [%, (n)] | 46.7 (14) | 9.1 (1) | 68.4 (13) | – | – |
| **General anesthetic management** | | | | | |
| Nitrous oxide for induction [%, (n)]* | 76.7 (23) | 54.5 (6) | 94.7 (17) | – | 0.07 |
| Propofol [%, (n)]* | 36.7 (11) | 27.3 (3) | 42.1 (8) | – | 0.47 |
| Propofol cumulative dose (mg/kg)* | 15 (CI: 10–20) | 10 (CI: 10-10) | 15 (CI: 10–20) | 5 | 0.08 |
| Median duration of anesthesia (min)* | 108.5 (CI: 87–145) | 118.0 (CI: 81–268) | 94.0 (CI: 76–160) | −24.0 | 0.18 |
| **MOSSA epoch detail** | | | | | |
| End-tidal sevoflurane (%)* | 2.6 (CI: 2.2–2.7) | 2.0 (CI: 0.8–2.6) | 2.6 (CI: 2.4–3.1) | 0.7 | **0.002** |

Data given as median with 95% Confidence Interval (CI) Limit, unless otherwise stated. All infants included in MOSSA analysis. Six infants were excluded from the emergence analysis due to anesthetic management or technical reasons (**Figure 2**); 3 infants aged 0–3 months and 3 infants aged 4–6 months. 95% CI limit of median.

MOSSA, Maintenance Of a Surgical State of Anesthesia.

*Mann–Whitney U-test.

†Fisher's exact test.

p < 0.05 considered statistically significant. **Supplementary file 1** provides characteristics for individual infants included in the analysis.

## EEG features during MOSSA

### Slow and delta oscillations are present from 0 to 6 months of age; theta and alpha oscillations emerge around 4 months of age

Frontal power spectra analysis in individual infants at F7 showed that slow (0.1–1 Hz) and delta power (1–4 Hz) were present at all ages (**Figure 3**). Power in the 4–30 Hz frequency range was minimal in infants at 1, 2, and 3 months of age (**Figure 3A,B**). Power in the theta (4–8 Hz) and alpha (8–12 Hz) frequency ranges emerged at 4, 5, and 6 months of age (**Figure 3C,D**). Based on these key features of the individual power spectra, infants were divided according to postnatal age into two groups, 0–3 months (n = 11) and 4–6 months (n = 19) for subsequent analyses.

Frontal group-median spectrograms computed at F7 in infants 0–3 months and 4–6 months of age show that power in the slow and delta ranges was dominant over that in other frequencies in both age groups (**Figure 3B,D**). Oscillations in the theta and alpha ranges were more prominent in infants aged 4–6 months during MOSSA.

Power across all frequencies was lower in infants at 0–3 months compared to infants at 4–6 months of age during MOSSA (**Figure 4A**). Infants 0–3 months of age showed negligible frontal alpha power (F7) with a peak of −6.52 dB at 8 Hz. Infants 4–6 months of age showed increased alpha power with a peak of 6.68 dB at 9 Hz. Group comparisons showed that power was significantly lower (95% Confidence Interval (CI), bootstrap analysis) across all of the frequencies from 0 to 30 Hz in infants 0–3 months of age, when compared with infants 4–6 months of age during MOSSA (**Figure 4B–E**).

End-tidal sevoflurane requirements during MOSSA were significantly lower in infants 0–3 months compared to infants 4–6 months of age (**Table 1**). Therefore, we performed a secondary analysis to control for potential dose effects by comparing EEG spectra during MOSSA at identical end-tidal

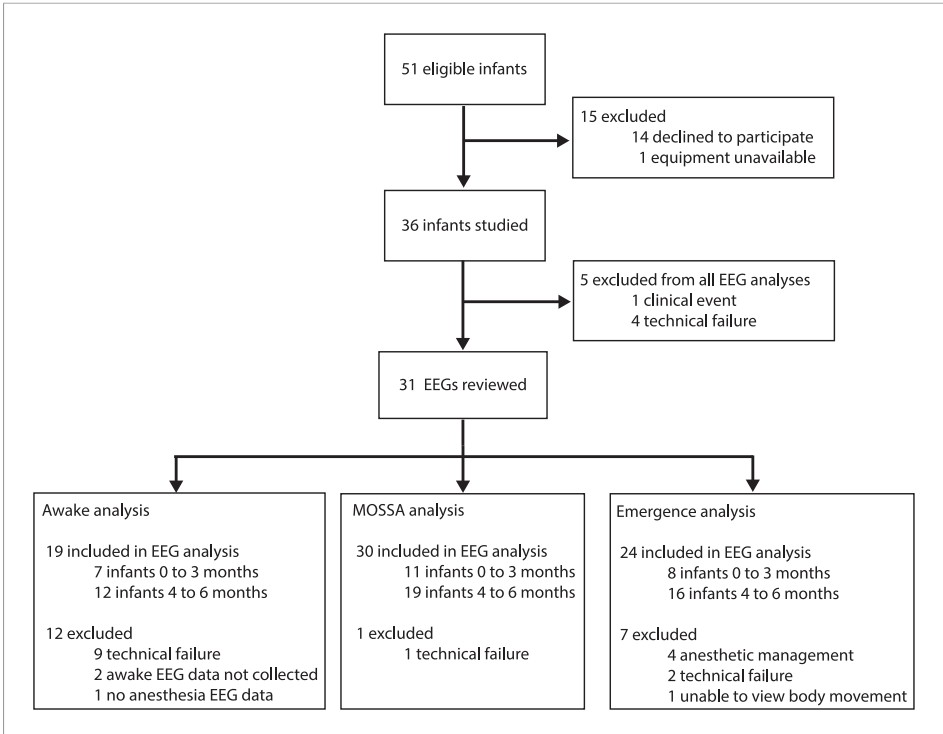

**Figure 2**. Study profile. Parents of 51 infants were approached and 36 consented. Five EEG recordings were excluded from the final analysis because of technical failures (n = 4) or a clinical event (n = 1; local anesthetic toxicity prior to surgical incision). For MOSSA analysis, data are presented from 30 EEG recordings (0–3 months, n = 11; 4–6 months, n = 19); for awake analysis, 19 infants were included (0–3 months, n = 7; 4–6 months, n = 12); and for emergence from general anesthesia analysis, 24 infants were included (0–3 months, n = 8; 4–6 months, n = 16).

sevoflurane concentrations of 1.8% across infants. We chose 1.8% end-tidal sevoflurane because all the infants received this concentration for at least 30 s during surgery and did not exhibit body movement or reflex activity at this plane of anesthesia suggesting a behavioral state comparable to MOSSA. At 1.8% end-tidal sevoflurane, frontal group-median spectrograms at F7 show that slow and delta power was dominant over other frequencies in both age groups (*Figure 4—figure supplement 1A,B*). Higher frequency oscillations in the theta and alpha range were more prominent in infants 4–6 months of age (*Figure 4—figure supplement 1C–G*). The patterns of EEG activity observed during MOSSA were also seen during lighter planes of MOSSA at 1.8% end-tidal sevoflurane suggesting that the observed differences between the two groups are not likely due to a dose effect.

## Frontal alpha predominance is low at 0–6 months of age

Group-median spectrograms were computed for each recording electrode location in the EEG montage (*Figure 5*). Infants 0–3 months of age showed slow and delta power at each electrode location and an absence of appreciable power at any frequency 5 Hz and higher. The power difference between the slow–delta range and the >5 Hz range was 30–35 dB. Infants 4–6 months of age group showed appreciable power in the slow, delta, theta, and alpha bands in the parietal, temporal, and non-central electrode locations; all the midline electrode locations showed less power in the frequencies 5 Hz and higher.

Topographic maps of power were computed for slow, delta, theta, alpha, and beta frequency bands (*Figure 6*). For both age groups, spectral power along the midline (specifically frontal and central-parietal locations) was lower across all frequencies (*Figure 6A,B*). Slow and delta oscillations were broadly distributed across the entire scalp, with slightly lower power located over the frontal, centro–parietal, and occipital midline compared to the temporal regions in all infants. Theta activity

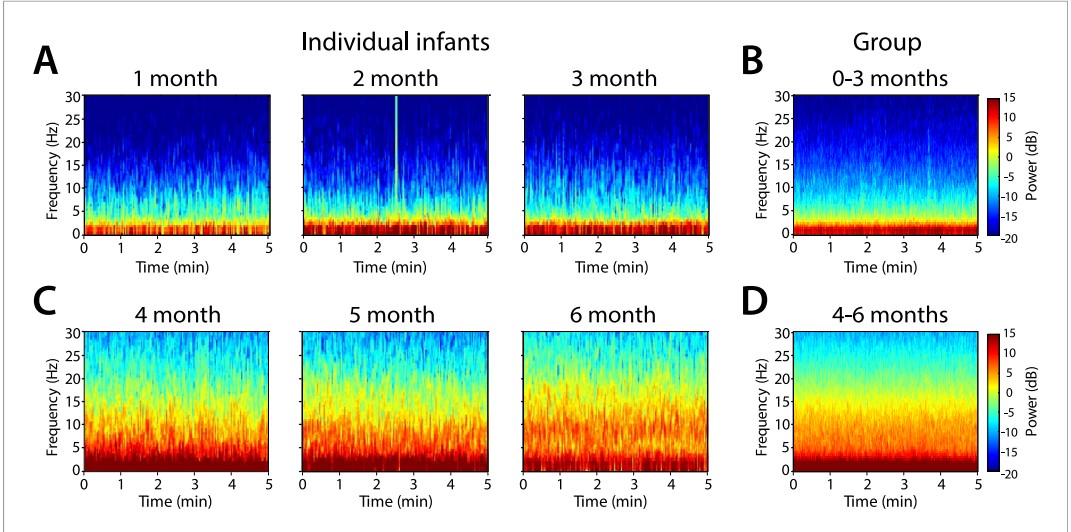

**Figure 3**. Frontal spectrograms during MOSSA. Individual infant EEG spectral power for frequencies from 0 to 30 Hz (left hand axis) during a 5-min period of MOSSA. Frontal spectrograms for infants at 0–3 months of age are shown for (**A**) individual infants and (**B**) group-median average. Frontal spectrograms for infants at 4–6 months of age are shown for (**C**) individual infants and (**D**) group-median average. F7 used with nearest neighbor Laplacian referencing.

followed a similar distribution pattern in all infants, although infants 0–3 months of age had minimal power in this range (<0 dB) (*Figure 6A*). During MOSSA, alpha oscillations, compared to lower frequency bands, were distributed more laterally across the frontal, central, and parietal regions in infants 4–6 months of age (*Figure 6B*). Alpha power was slightly lower over the central–parietal and occipital midline, possibly reflecting changes in skull conductivity over the locations of fontanels, rather than an artifact of referencing technique.

We evaluated the potential frontal predominance of alpha power by assessing power spectra differences between frontal (FPz) and occipital (Oz) electrodes for infants 0–3 months of age (*Figure 7A,C–F*) and for infants 4–6 months of age (*Figure 7B,G–J*). During MOSSA, there were no significant frontal-occipital power differences at any frequency in infants at 0–3 months (95% CI, paired bootstrap analysis) (*Figure 7C–F*). In contrast, a small but significant increase in frontal alpha power compared to occipital power (95% CI, paired bootstrap analysis) was present during MOSSA in infants 4–6 months of age (*Figure 7G–J*). These data suggest, that unlike in adults, infants 0–3 months of age do not show frontal predominance of alpha power during MOSSA, and that it may begin to emerge at around 4 months postnatal age (*Ní Mhuircheartaigh et al., 2013*; *Purdon et al., 2013*).

## Frontal alpha coherence is absent at 0–6 months of age

We analyzed the level of local coordinated activity in the frontal regions of the scalp during MOSSA by computing the coherogram (time-varying coherence) and coherence between the left (F7) and right (F8) frontal electrodes (see 'Materials and methods') (*Figure 8A–D*). The frontal coherograms showed high coherence (~0.6) in the slow and delta frequency ranges for all infants during MOSSA (*Figure 8E,F*). In infants 0–3 months of age, the peak coherence was 0.61 and occurred at 1.0 Hz (*Figure 8G*). In infants aged 4–6 months, peak coherence was also 0.61, and occurred at 1.0 Hz (*Figure 8G*). These results suggest that infants across all postnatal ages have coordinated local frontal slow and delta oscillations during MOSSA.

Although infants 4–6 months of age showed strong power in the theta and alpha bands (*Figure 8C,D*), oscillations in these bands were not coherent (*Figure 8G*). This contrasts with adults who show a strong frontal coherence in the alpha band during sevoflurane general anesthesia (*Akeju et al., 2014c*). There were no discernable differences between age groups in the level of coherence between 0.1 and 30 Hz (95% CI, bootstrap analysis).

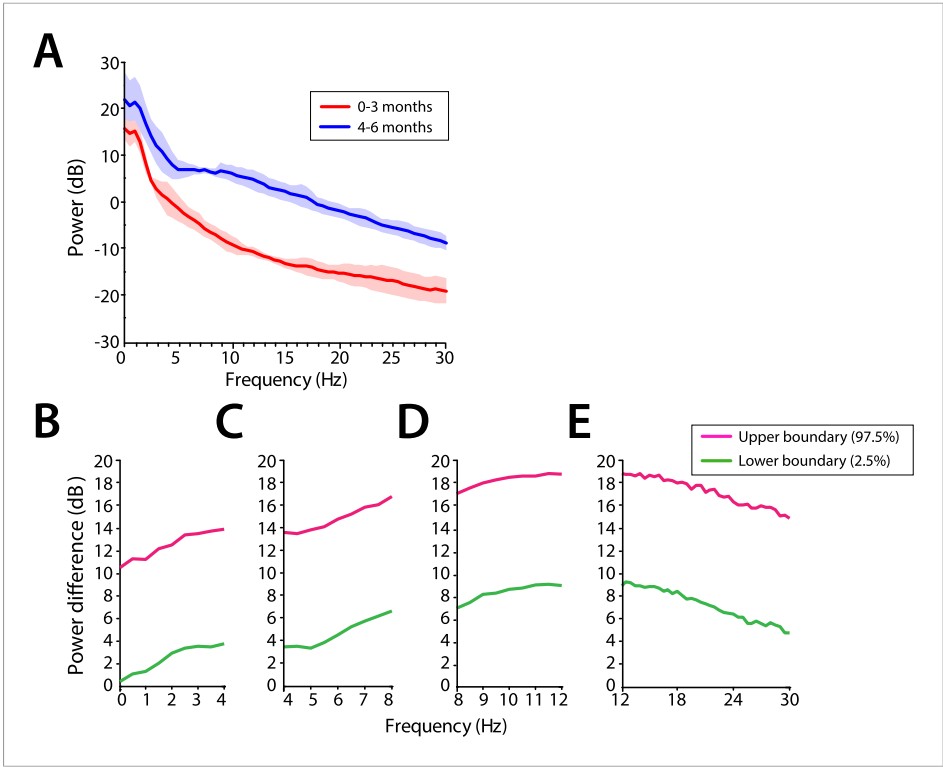

**Figure 4**. Frontal EEG power is greater in infants 4–6 months of age across all frequencies during MOSSA. (**A**) Frontal group-median power spectra (solid line, median; shaded area, 25th–75th percentile) across the 0–30 Hz-frequency band showing increased power in infants 4–6 months of age across all frequencies. (**B-E**) Differences in frontal group-median power spectra presented with 95% CI from bootstrap analysis (pink line, 97.5th percentile; green line, 2.5th percentile) comparing infants at 0–3 month to 4–6 months of age. (**B**) Power difference in slow and delta frequency range, 0–4 Hz, (**C**) theta, 4–8 Hz, (**D**) alpha, 8–12 Hz, and (**E**) beta, 12–30 Hz. F7 electrode presented using nearest neighbor Laplacian referencing.

The following figure supplement is available for figure 4:

**Figure supplement 1**. Frontal EEG spectral properties in infants anesthetized under a uniform end-tidal sevoflurance concentration (1.8%).

## Global coherence is weak across all frequencies at 0–6 months of age

Global coherence analysis characterizes coordinated activity across multiple channels of EEG as a function of frequency. Global coherence is the fraction of variance at a given frequency across all EEG channels that is explained by the first eigenvector of the cross-spectral matrix. Topographic maps of coherence were computed for slow, delta, theta, alpha, and beta frequency bands in infants 0–3 months of age (*Figure 9A*) and infants 4–6 months of age (*Figure 9B*) during MOSSA. Despite slow and delta oscillations in the spectrograms at all ages, and theta and alpha oscillations in the spectrograms of infants 4–6 months of age, global coherence was weak (<0.2) in all frequency bands for all infants (*Figure 9C*). This finding contrasts with the pattern of highly coordinated, coherent frontal alpha activity observed in adults during propofol general anesthesia (*Cimenser et al., 2011*; *Purdon et al., 2013*; *Akeju et al., 2014c*). The lack of global coherence in the slow and delta frequency ranges is consistent with the lack of slow oscillation coherence observed in adults (*Cimenser et al., 2011*; *Lewis et al., 2012*; *Purdon et al., 2013*).

## EEG features during awake state

We evaluated the EEG features during the awake state (prior to anesthesia exposure) in infants whom awake EEG could be recorded. This was 7 infants in the 0–3 month age group and 12 infants in the

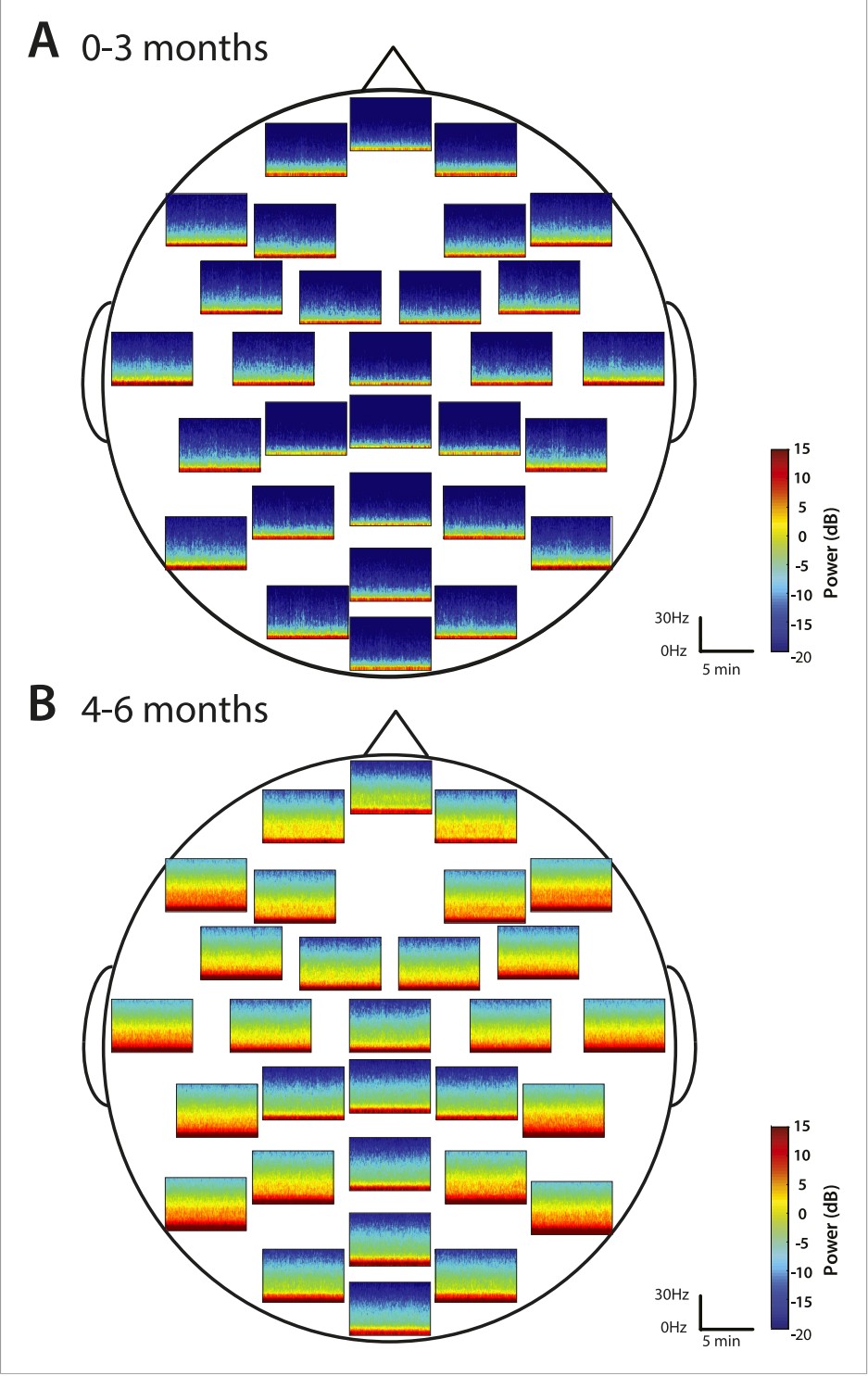

**Figure 5**. Spatial distribution of spectral power during MOSSA. Group-median spectrograms at each recording electrode location across the scalp in infants at (**A**) 0–3 months (n = 11) and (**B**) 4–6 months (n = 19).

4–6 month age group. Body movement and/or eye opening prior to exposure of sevoflurane general anesthesia were used to confirm the presence of the awake state. Frontal group-median spectrograms at F7 showed dominant slow and delta power, with low theta and alpha power in all ages, were the predominant frequencies during the awake state at all ages (*Figure 10—figure supplement 1A–C*).

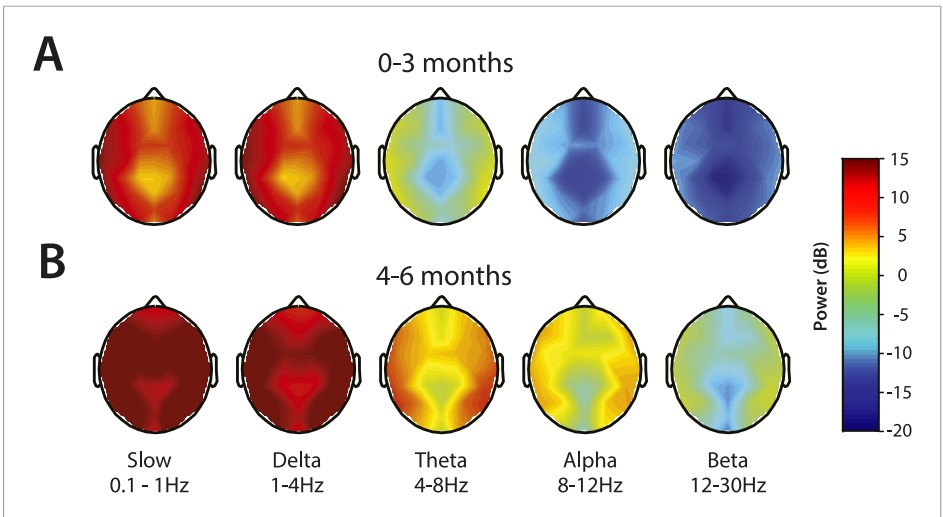

**Figure 6**. Topographic EEG maps of spectral power for distinct frequency bands during MOSSA. Topographic EEG maps detailing group-averaged power for each EEG frequency band in infants aged (**A**) 0–3 months (n = 11) and (**B**) 4–6 months (n = 19). Slow-wave activity is distributed across the scalp in both postnatal age groups, while alpha activity is present to a greater degree in infants 4–6 months postnatal age. Legend for **A** and **B** is shown by the color bar.

Power across all frequencies was significantly lower in infants at 0–3 months compared to infants at 4–6 months of age (95% CI, bootstrap analysis; *Figure 10—figure supplement 1D–G*).

## Comparison of EEG spectral features in the awake (pre-anesthesia) state and MOSSA

### Theta and alpha oscillations increase in power during MOSSA in infants 4–6 months of age

Slow and delta power remained relatively constant in all infants during the awake state and MOSSA (*Figure 10A,B*). There were no clear associations between power in any frequency band and the awake state or MOSSA in infants 0–3 months of age (*Figure 10A*). Power in the theta (>5 Hz) and alpha frequency ranges were prominent features during MOSSA in infants 4–6 months of age (*Figure 10B*). Infants 0–3 months of age showed no change in theta or alpha activity associated with general anesthesia (95% CI paired bootstrap analysis, *Figure 10D,E*). In infants 4–6 months of age, the relationship between theta and alpha power and general anesthesia was significant (95% CI, paired bootstrap analysis; *Figure 10H,I*).

## EEG features during emergence

### Theta and alpha oscillations decrease in power during emergence in infants 4–6 months of age

On emergence, infants started to display gross body movement at a median end-tidal concentration of 0.6% (95% CI, 0.2–1.0%), and 0.4% (95% CI, 0.3–0.6%) at 0–3 month of age and 4–6 months of age, respectively (*Figure 11*). No infants exhibited body movement at 1.2% or greater end-tidal sevoflurane concentrations.

Slow and delta power remained relatively constant in infants of all ages as end-tidal sevoflurane concentration decreased during MOSSA through emergence (*Figure 11*). There were no clear associations between power in any frequency band and change in end-tidal sevoflurane concentration in infants 0–3 months of age (*Figure 11A*). Theta and alpha power was constant and sizable at end-tidal sevoflurane concentrations greater than 1.2% in infants 4–6 months of age (*Figure 11B*). Theta and alpha power began to decrease once end-tidal sevoflurane concentration fell below 1.2%, and gross body movement started to appear, in infants 4–6 months of age. In these older infants, theta and alpha activity decreased to negligible power levels at end-tidal sevoflurane concentrations where

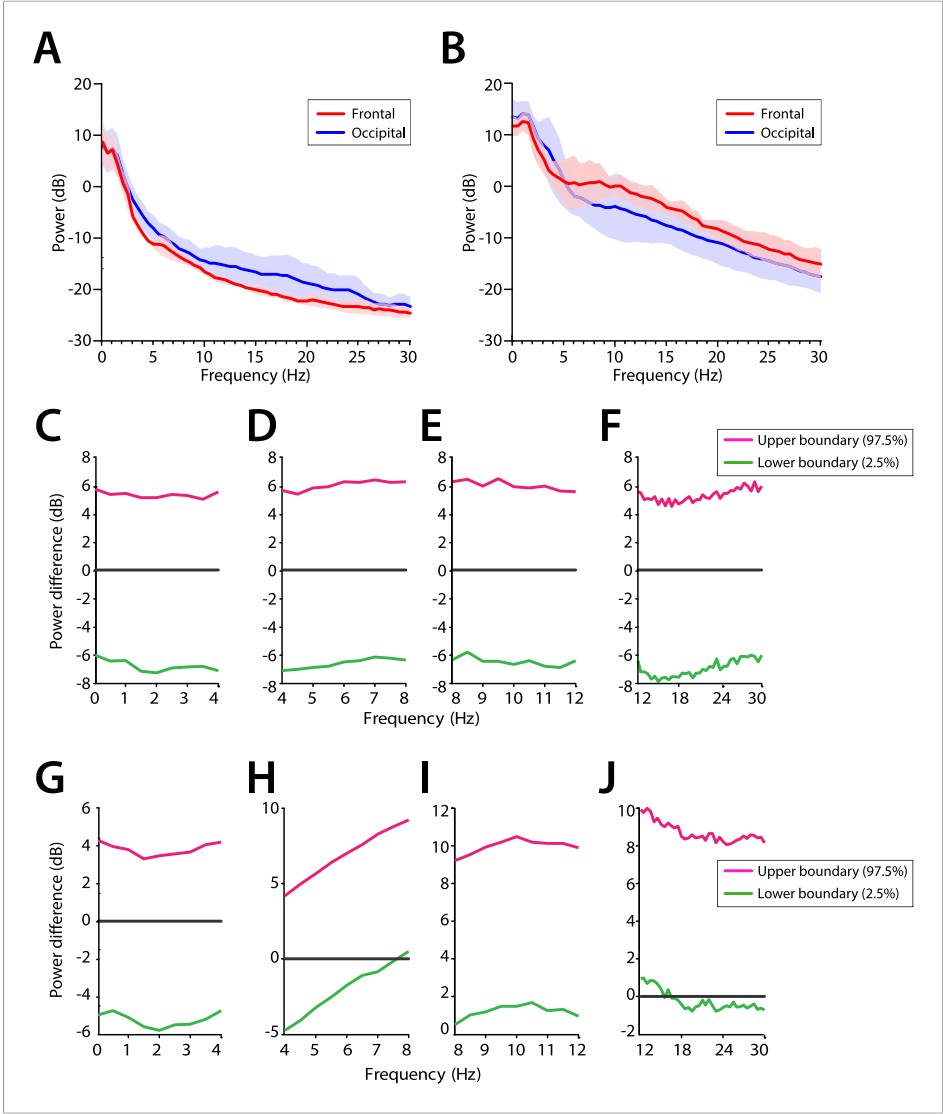

**Figure 7**. Frontal predominance of alpha power is low during MOSSA in infants 0–6 months of age. Frontal group-median power spectra (solid line, median; shaded area, 25th–75th percentile) showing similar EEG power across all frequencies in frontal and occipital channels in infants aged (**A**) 0–3 months (n = 11) and (**B**) 4–6 months (n = 19). Differences in frontal group-median power spectra presented with 95% CI from bootstrap analysis (pink line, 97.5th percentile; green line, 2.5th percentile) between frontal and occipital channels during MOSSA in infants aged (**C–F**) 0–3 months and (**G–J**) 4–6 months. A small but significant increase in frontal alpha power compared to occipital alpha power begins to emerge at 4–6 months of age. FPz and Oz electrodes presented using nearest neighbor Laplacian referencing.

more than 50% of infants displayed gross body movement. Topographic mapping of the power spectral content showed that there was no spatial shift in alpha activity at any end-tidal sevoflurane concentration in older infants. This finding contrasts the adult pattern where anesthesia-induced frontal alpha oscillations disappear and are replaced by posterior alpha during recovery of consciousness (*Ní Mhuircheartaigh et al., 2013*; *Purdon et al., 2013*).

## Comparison of EEG spectral features during MOSSA and first body movement

We evaluated frontal power spectra at the frontal channel (F7) at two time periods (1) MOSSA, and (2) immediately after first body movements were recorded (*Figure 12*). Infants 0–3 months of age showed no change in power across any frequency with recovery of movement during emergence

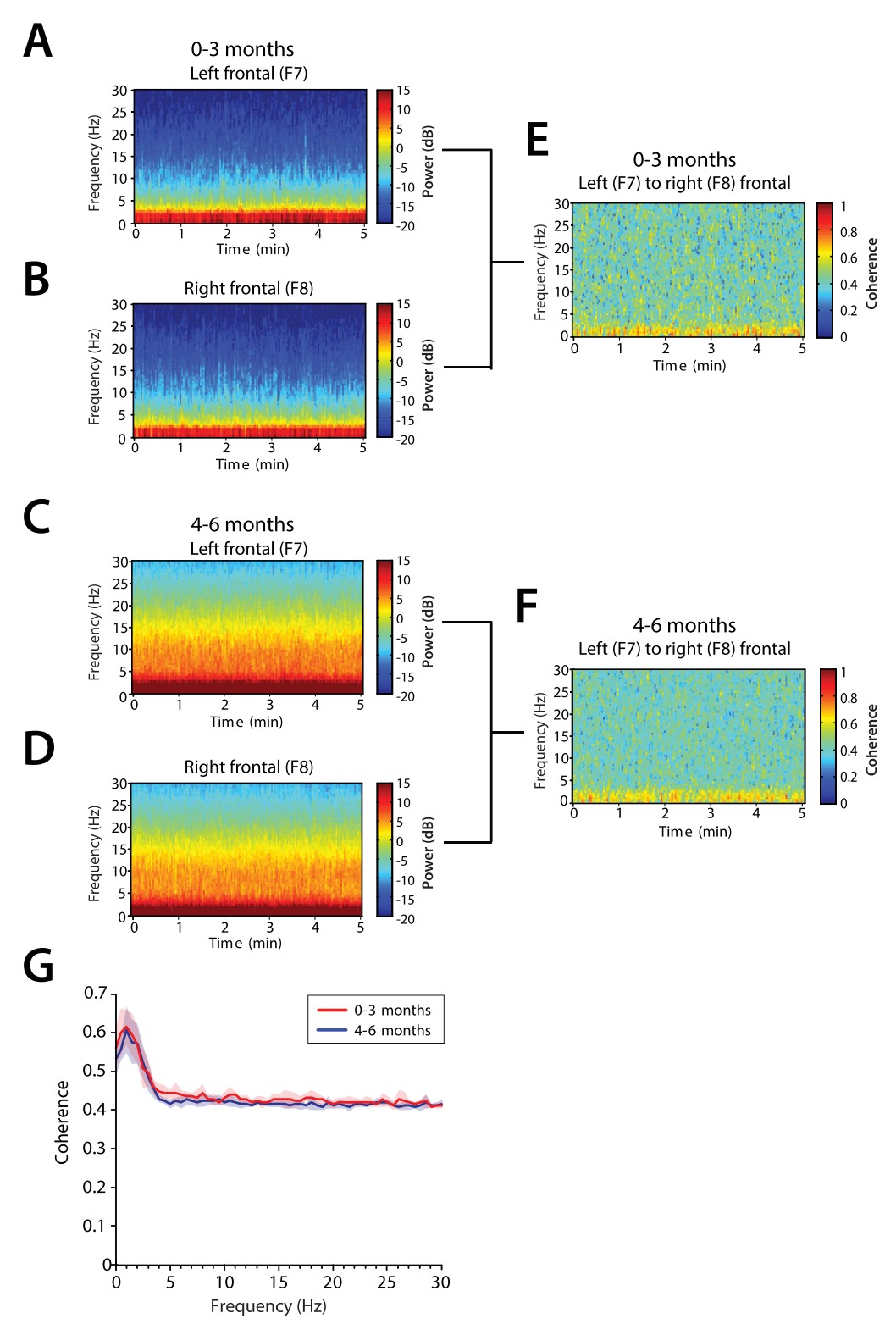

**Figure 8**. Frontal alpha coherence is absent during MOSSA in infants from 0 to 6 months postnatal age. Group-averaged frontal spectrograms in infants aged 0–3 months at (**A**) Left—F7, and (**B**) Right—F8, and infants aged 4–6 months (**C**) F7, and (**D**) F8. Relative group-averaged frontal coherogram (F7–F8) for infants aged (**E**) 0–3 months and (**F**) 4–6 months age. (**G**) Frontal group-median coherence (solid line, median) showed similar coherence across 0.1–30 Hz frequency bands at all postnatal ages.

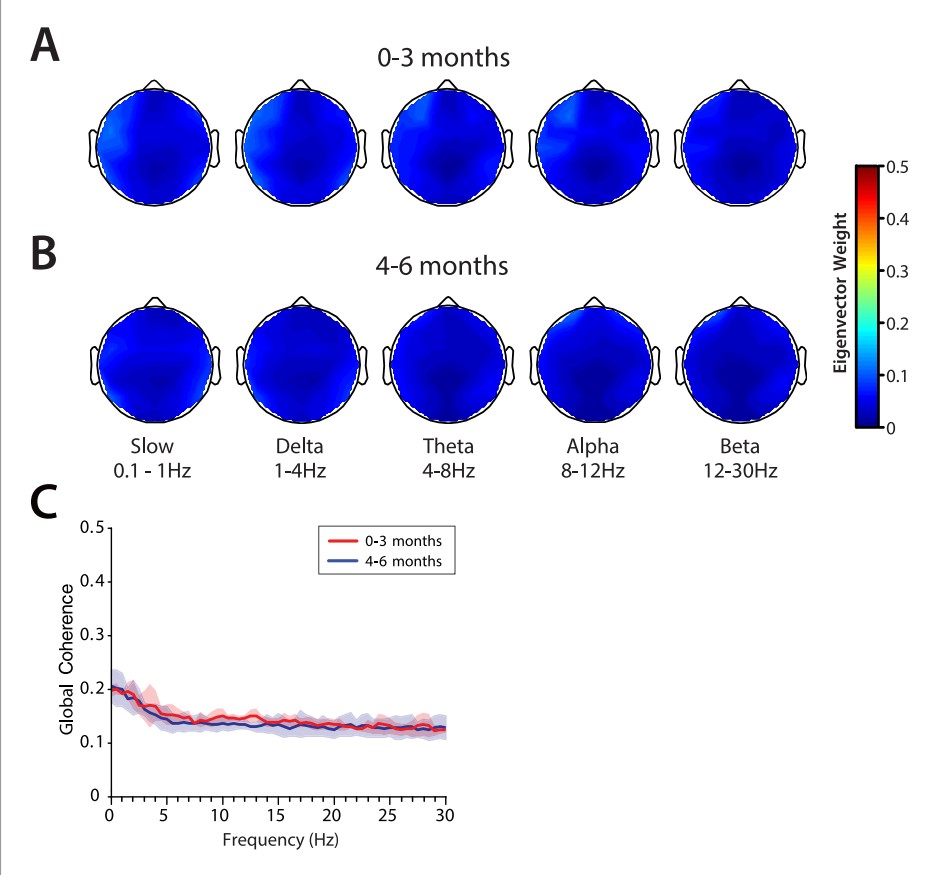

**Figure 9**. Global coherence is low across all frequencies during MOSSA in infants from 0 to 6 months of age. Topographic EEG maps detailing group-averaged global coherence for each EEG frequency band in infants (**A**) 0–3 months (n = 11) and (**B**) 4–6 months (n = 19) of age. (**C**) Group-median global coherence spectra (solid line, median; shaded area, IQR) show similar low coherence across 0–30 Hz frequency bands at all ages during MOSSA.

(95% CI, paired bootstrap analysis; *Figure 12C–F*). In infants 4–6 months of age, the relationship between theta and alpha power and recovery of movement was significant (95% CI, paired bootstrap analysis; *Figure 12G–J*). We note that two infants included in the 0–3 month age group exhibited low levels of alpha power at MOSSA that decreased during emergence. These infants were 107 and 116 postnatal days, almost 4 months of age, and suggest the underlying circuitry begins to become more refined at this developmental age, starting to reflect similar properties to infants 4–6 months of age.

## Discussion

We used multi-electrode EEG recordings to characterize the spatial and temporal dynamics of brain activity in infants 0–6 months of age during the awake state, and maintenance of and emergence from sevoflurane general anesthesia for routine surgical care. We showed that during maintenance: (1) all infants had slow oscillations across the entire scalp (*Figures 3–6*); (2) theta and alpha oscillations were present in addition to the slow oscillations across the entire scalp in all infants 4 months and older (*Figures 3–6*); (3) across all frequencies, the frontal power in the EEG was significantly greater in the 4- to 6-month-old infants compared to 0- to 3-month-old infants (*Figure 4*); (4) unlike in adults, all 0- to 6-month-old infants showed an absence of frontal alpha coherence as well as an absence of frontal predominance in power (*Figures 5–9*), although a small but significant increase in frontal alpha power compared to occipital alpha power emerged in infants 4–6 months of age (*Figure 6*). We also showed that alpha oscillations increase in power when switching from the awake state to MOSSA in infants 4–6 months of age (*Figure 10*); and that during emergence, in infants 4–6 months of age power decreased

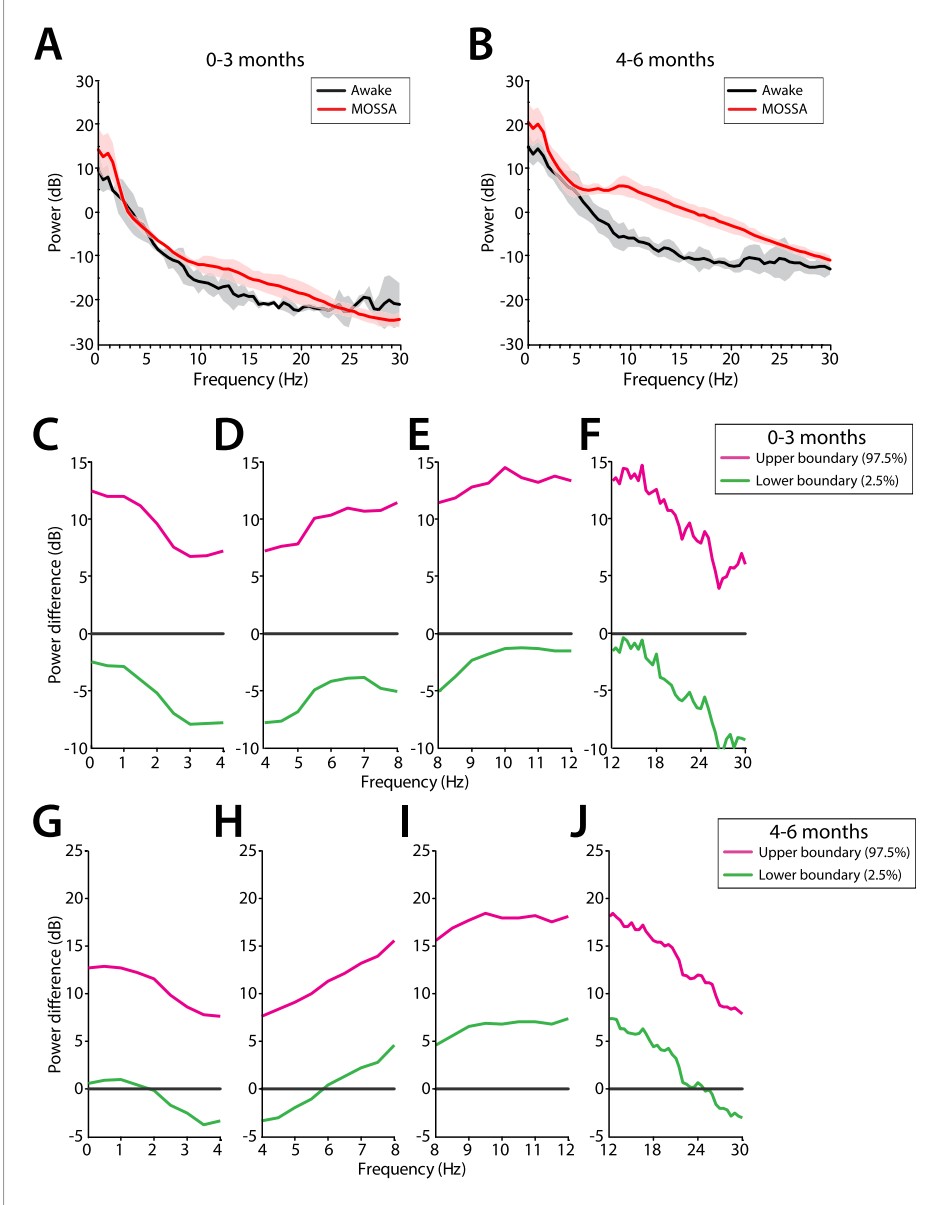

Figure 10. Frontal EEG power changes between the awake state and MOSSA. (**A**) Group-averaged frontal power spectra (solid line, median; shaded area, 25th–75th percentile) show similar EEG power during the awake state (prior to anesthesia) and MOSSA in infants aged 0–3 months across all frequencies (n = 7). (**B**) Group-averaged frontal power spectra show increased theta, alpha, and gamma oscillations during MOSSA in infants 4–6 months of age (n = 12). Differences in group-averaged frontal power spectra presented with 95% CI from paired bootstrap analysis (pink line, 97.5th percentile; green line, 2.5th percentile) between Awake and MOSSA in infants (**C**–**F**) 0–3 months and (**G**–**J**) 4–6 months of age. F7 electrode presented using nearest neighbor Laplacian referencing. Epoch size of 11 s is used for awake and MOSSA states.

The following figure supplement is available for figure 10:

**Figure supplement 1**. Frontal EEG spectral properties in awake infants.

in the alpha and theta bands with decreasing end-tidal sevoflurane concentration (*Figures 11*, *12*). Because we made video recordings during our studies, we related changes in anesthetic concentration to return of movement as an approximate behavioral marker of emergence from general anesthesia (*Figures 11*, *12*).

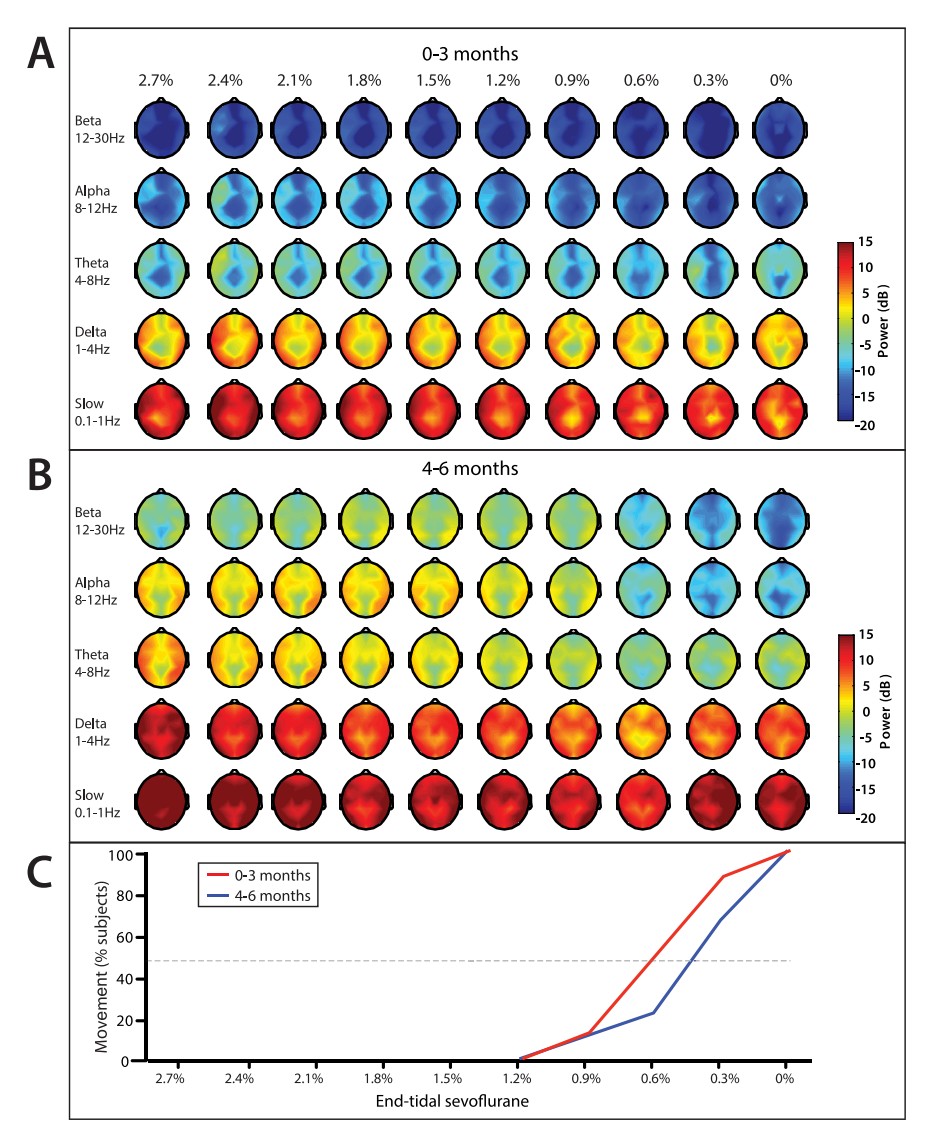

**Figure 11**. End-tidal sevoflurane concentration associated spatial EEG power and body movement during emergence from sevoflurane general anesthesia. (**A**) End-tidal sevoflurane concentration increments from MOSSA to emergence from general anesthesia for infants 0–3 months (n = 8), and (**B**) 4–6 months of age (n = 16), and (**C**) the corresponding percentage of infants who displayed gross body movement MOSSA.

## Studies of EEG dynamics of infant brain under general anesthesia

Age-related differences in EEG properties have been shown in previous studies of infants and children receiving general anesthesia. Davidson and colleagues monitored 8 leads of EEG in 17 infants, 0–6 months of age, who received either sevoflurane or propofol general anesthesia (*Davidson et al., 2008*). These infants were part of a study cohort that ranged in age from 9 days to 12 years. While total spectral power—limited to 2–20 Hz—did not differ between the anesthetized state and emergence, the EEG traces of the infants during emergence showed bursts of electrical activity interspersed with periods of low-amplitude activity. In contrast, the 21 children aged 2–10 years showed a significant increase in SEF90 and a significant decrease in total frontal EEG power during emergence. Lo and colleagues studied emergence by monitoring 128 leads of EEG in children 22 days to 3.6 years who received either sevoflurane or isoflurane anesthesia (*Lo et al., 2009*). These investigators reported differences between the spatial EEG patterns of the two agents. However, none of the analyses was stratified by age.

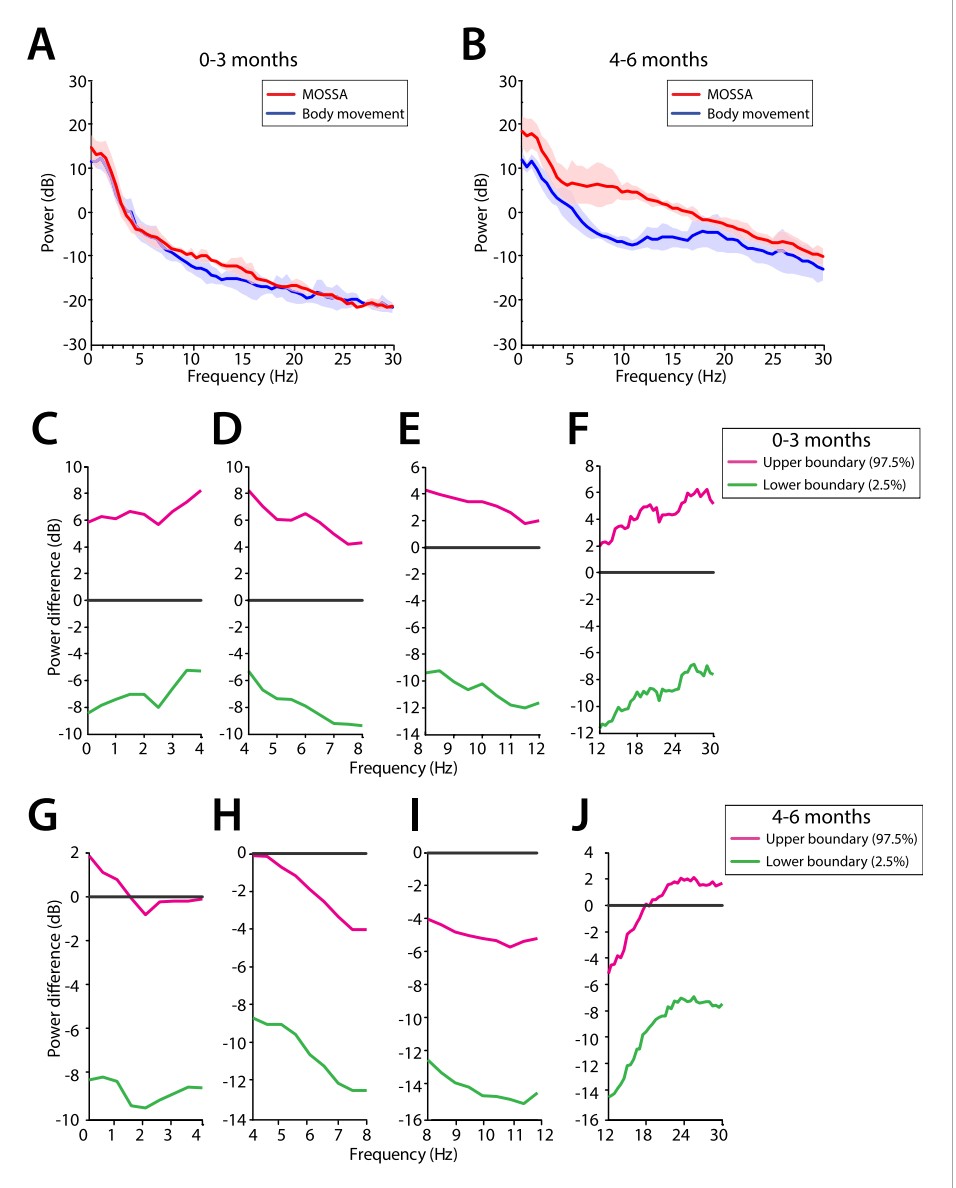

**Figure 12**. Frontal EEG power changes between MOSSA and body movement. (**A**) Frontal group-median power spectra (solid line, median; shaded area, 25th–75th percentile) show similar EEG power during MOSSA and after first body movement in infants aged 0–3 months across all frequencies. (**B**) Frontal group-median power spectra show increased theta and alpha oscillations during MOSSA in infants aged 4–6 months. Differences in frontal group-median power spectra presented with 95% CI from bootstrap analysis (pink line, 97.5th percentile; green line, 2.5th percentile) between MOSSA and emergence after first body movement in infants (**C**–**F**) 0–3 months and (**G**–**J**) 4–6 months of age. F7 electrode presented using nearest neighbor Laplacian referencing.

Hayashi and colleagues conducted a retrospective study on EEG data collected using a 2-electrode montage in 62 neonates and infants ranging in age from 1 day to 2 years who received sevoflurane general anesthesia (*Hayashi et al., 2012*). They analyzed changes in 90% spectral edge frequency (SEF90), burst suppression ratio (BSR), relative beta ratio (RBR), and approximate entropy (ApEn) to assess changes in brain activity during emergence. They found that infants less than 6 months had little to no change in EEG parameters during emergence, whereas the infants 6 months and older showed the greatest changes in the EEG parameters, which were an increase in the SEF90, decrease in the BSR, increase in the RBR, and decrease in ApEn. The infants in the range of 3–5 month old

marked the transition between no EEG changes on emergence to highly visible changes on emergence. Sury and colleagues studied 20 infants at 1 week to 10 months of age during emergence from sevoflurane general anesthesia using a 4-lead EEG montage to analyze spectral power between 1 and 28 Hz (*Sury et al., 2014*). They found that infants 3 months and older had greater power in the 5–20 Hz range during maintenance than infants less than 3 months and that the power in the older infants decreased on emergence.

Our results refine the results in these reports by showing that there are readily discernible differences in the anesthetic responses of infant 0–3 months of age and infants 4–6 months of age. We report for the first time detailed spatio-temporal analyses of infant 33-lead EEGs with Laplacian referencing using spectra (analyzed from 0.1–30 Hz), spectrograms, coherograms, and global coherence computed by multitaper spectral analysis methods. These methods, which are known to have several important optimality properties, have been used successfully to characterize the EEG dynamics of adults receiving general anesthesia and sedation (*Cimenser et al., 2011*; *Ní Mhuircheartaigh et al., 2013*; *Purdon et al., 2013*; *Akeju et al., 2014b, 2014c*). Moreover, the spectrograms are particularly important because they are the frequency domain representations of the EEG displayed on commercially available EEG monitors.

## Studies of EEG dynamics in adults under general anesthesia

It is now appreciated that in adults at surgical levels of general anesthesia maintained by GABAergic anesthetics (ether-based agents, barbiturates, propofol), the EEG shows slow oscillations and alpha oscillations, in addition to anteriorization, defined as the movement predominantly of the EEG power in the alpha band from the occipital to the frontal area of the head (*Feshchenko et al., 2004*; *Cimenser et al., 2011*; *Murphy et al., 2011*; *Ní Mhuircheartaigh et al., 2013*; *Purdon et al., 2013*; *Akeju et al., 2014b, 2014c*). Studies of propofol have shown that these alpha oscillations are strongly coherent across the front of the head during unconsciousness, whereas the slow oscillations are not coherent anywhere across the head (*Lewis et al., 2012*; *Purdon et al., 2013*). With return of consciousness, the frontal alpha power disappears, and posterior alpha reappears (*Purdon et al., 2013*). Because we did not record EEG in the infants continuously and then administer sevoflurane in a gradual escalating and a gradual deescalating manner, our study could not assess anteriorization. During MOSSA, we see a small, but significant, increase in frontal alpha power when compared to occipital power in infants 4–6 months, which suggests this feature of anesthetic-induced consciousness emerges in early life. Nevertheless, our findings demonstrate that the spatio-temporal dynamics of the anesthetized states of 0- to 3-month-old and 4- to 6-month-old infants differ appreciably from each other and from those of adults.

## Mechanisms for alpha, slow, and delta oscillations

Recent modeling studies of propofol offer insights into the possible origins of these differences. These studies suggest that the frontal EEG alpha oscillations likely represent oscillatory activity between the cortex and thalamus (*Ching et al., 2010*). Other modeling work suggests that anteriorization is due to different effects of propofol on thalamic nuclei that project to occipital areas compared to thalamic nuclei that project to areas of the pre-frontal cortex (*Vijayan et al., 2013*). Propofol disrupts the normal, depolarized alpha oscillations in posterior-projecting thalamic nuclei whereas it induces, hyperpolarized alpha oscillations in frontal-projecting thalamic nuclei. The differential effect appears to be due primarily to propofol's inhibition of subset of posterior-projecting neurons with hyperpolarizing activating $I_h$ currents that typically produce alpha oscillations at resting membrane voltages above 60 millivolts (*Vijayan et al., 2013*). The slow and delta oscillations during general anesthesia are consistent with anesthetic-induced decreases in major excitatory brainstem inputs to the cortex. GABAergic anesthetic likely produce these effects by acting at the GABAergic synapses arising from inhibitory neurons in the pre-optic area of the hypothalamus that project onto each of the major arousal nuclei in the midbrain and upper pons (*Brown et al., 2011*). Both the alpha, slow and delta oscillations likely reflect disruption of thalamo-cortical and cortico-cortical processing required to maintain consciousness (*Ching et al., 2010*; *Purdon et al., 2013*).

The model predictions for the frontal predominance of alpha oscillations await experimental verification. However, the neuroanatomy and neurophysiology underlying the putative mechanism for alpha oscillations suggest that the thalamocortical connections required to produce the alpha

oscillations are absent in infants 0–3 months old yet, present in infants 4–6 months old. The putative mechanism for frontal predominance of alpha power suggests that the differential thalamic connectivity required to produce this phenomenon is not present in infants 6 months of age or younger.

## Brain development as a source of changes in EEG dynamics with age

Developmental changes in the brain are the obvious mechanisms to explain the changes in EEG dynamics that occur with age. Gross brain development occurs in a caudal to rostral direction, with myelination of the medulla, pons, and thalamus starting within the first few postnatal weeks and frontal cortex myelination starting around postnatal months 3–4 (*Brody et al., 1987*; *Kinney et al., 1988*). In addition, there is significant synaptogenesis, neuronal differentiation, and pruning, along with changes in GABAergic neurotransmission (*Huttenlocher and Dabholkar, 1997*; *Hensch, 2004*; *Tau and Peterson, 2010*; *Dehorter et al., 2012*; *Catts et al., 2013*; *Semple et al., 2013*). Distinct regional differences in the rate of synaptogenesis, glucose metabolism, and myelination across the cortex occur between subcortical and cortical regions, and between different regions of the cortex during the first 12 postnatal months in human infants. For example, in the visual cortex, there is a rapid burst of synapse formation between 3 and 4 months, and the maximum density is reached between 8 and 12 months (*Huttenlocher et al., 1982*). Synaptogenesis starts at the same time in the pre-frontal cortex, but the density of synapses increases much more slowly and does not peak until after the first 12 months (*Johnson, 2001*). Additionally, Positron Emission Tomography (PET) studies show that between 0 and 3 months postnatal age, glucose uptake is highest in the occipital parietal, and temporal cortices; and by 6–8 months, glucose uptake extends to the frontal cortex appearing with higher cortical function (*Chugani and Phelps, 1986*; *Chugani et al., 1987*; *Kinnala et al., 1996*). A key role in brain development is played by the subplate neurons, the first neurons generated in the cerebral cortex, which guide formation of thalamocortical connections (*Kanold and Luhmann, 2010*; *Kostović and Judas, 2010*). The subplate cells form the first functional connections and are a must for relaying early oscillatory activity in the developing brain (*Kanold and Luhmann, 2010*). To the extent that the alpha oscillations in the anesthetized brain are postulated to be produced by thalamocortical circuits, the appearance of the alpha oscillations at 4 months of age may suggest that an important developmental milestone has been reached in the processes guided by the subplate neurons.

## Study limitations and constraints

The conduct of research in children carries with it all the ethical obligations of adult research along with the additional obligation of not exposing children to risks beyond those associated with their routine medical care. As a consequence, observational studies of children receiving general anesthesia or sedation as part of routine diagnostic or therapeutic care are and will continue to be the principal approach to studying the neurophysiology of general anesthesia in children. It is crucial to plan carefully these studies and their subsequent analyses in order to maximize the information learned on this important topic. Anesthetic management was only standardized in our study to the extent that the staff at Boston Children's Hospital uses similar practices. The specific anesthetic management in each case was carried out by the attending anesthesiologist. Given the strength of the findings we report, a more systematic protocol would likely provide more evidence for our findings.

Because administration of general anesthesia is a high-risk human study, dose-titration protocols that are common in adults for studying controlled induction and emergence cannot be conducted in children. Therefore, we conducted our dose–response analysis (*Figure 11*), like those reported in previous studies, by observing changes in end-tidal sevoflurane concentration during emergence instead of by recording EEG while systematically changing the anesthetic dose during induction. In place of formal behavioral assessments of consciousness, which would not be feasible in children, we used movement recorded on video as an approximate behavioral marker of emergence from general anesthesia. The advantage of using body movement as a measure of emergence is that it is well-defined in children, and video recordings of body movement can be time-locked to the EEG recording for quantitative analysis. Relating onset of emergence to changes in respiration and heart rate, as well as blood pressure, also reflect effects of anesthesia but in a more complex manner being dependent on drug administration and clinical state of the patient. Similar to what we observed in adults, we found concentration- and behavior-dependent reductions in alpha and theta power during

emergence in 4- to 6-month-old infants. Unlike in adults, we did not observe any relationship between slow/delta power and anesthetic concentration or behavior. The differences in slow/delta oscillation dynamics between these infants and adults may be attributable to a number of factors, including mechanistic differences the two groups, low signal-to-noise in the slow/delta band due to patient movement, and the possibility that the infants remained sedated or unconscious despite recovering movement immediately following surgery. Our comparisons between the awake state, MOSSA and emergence indicate that state-specific EEG spectral properties begin to emerge at 4–6 months of age. They demonstrate the challenges faced when using EEG measures to evaluate anesthetic depth in younger infants (0–3 months). Our findings can be strengthened by studying sevoflurane general anesthesia in children across the entire pediatric age spectrum, and by conducting similar studies of other the anesthetics, for example, propofol, dexmedetomidine, and isoflurane, commonly used in children.

### Clinical implications and future directions

In summary, we have shown that infants 0–6 months of age have markedly different EEG patterns from each other and from adults under general anesthesia. These differences are likely due to differences in structural and functional aspects of cortico-cortical and thalamocortical connectivity, and help explain why EEG-based indices provide inaccurate measures of anesthetic states in children, especially during the first three months of life. We introduced the use of multitaper spectral methods in the analysis of pediatric EEG recordings to facilitate comparisons with adult analyses. We provide spectrograms to show how these brain dynamics appear on available EEG monitors.

The design of strategies to track the brain states of children receiving general anesthesia and sedation has not received the attention that this topic has received in adults. Systems neuroscience research that takes account of brain development will be required to accurately define anesthetic states for the entire pediatric age spectrum and to devise principled neurophysiological-based strategies for anesthetic dosing in older infants (>3 months) and children. Moreover, until the question of whether anesthetics are toxic to the developing brain is answered, design of neurophysiological-based definitions of anesthetic states and design of neurophysiological-based brain monitoring strategies offer the most prudent approaches to mitigating anesthetic risk in this vulnerable population.

## Materials and methods

### Study design

The objective of this observational study was to evaluate the effect of postnatal age on electroencephalographic (EEG) activity during sevoflurane general anesthesia in infants 0–6 months old. We recorded multichannel EEGs during administration of sevoflurane general anesthesia for elective surgery, per clinical protocol. End-tidal anesthetic gas volume and video recordings of behavioral activity were time-locked to the EEG recording. The spatial and temporal properties of the infant EEG were evaluated during the awake state, and at two distinct periods during administration of sevoflurane general anesthesia: (1) MOSSA and (2) emergence from sevoflurane general anesthesia (Figure 1). The age-dependent effects of sevoflurane general anesthesia on the EEG were compared in two groups of infants who were (i) 0–3 months and (ii) 4–6 months of age.

### Participants

Infants who were scheduled for an elective surgical procedure were recruited from the pre-operative clinic at Boston Children's Hospital from December 2011 to August 2014. Eligibility criteria consisted of infants between 0 and 6 months postnatal age who required surgery below the neck. All infants were clinically stable on the day of study and American Society of Anesthesiologists' physical status I or II. Infants were not eligible for inclusion in the study if they were (1) born with congenital malformations or other genetic conditions thought to influence brain development, (2) diagnosed with a neurological or cardiovascular disorder, or (3) born at <32 weeks post-menstrual age.

Ethical approval was obtained from Boston Children's Hospital Institutional Review Board (Protocol Number IRB-P000003544) and classified as a 'no more than Minimal Risk' study. Informed written consent was obtained from parents/legal guardians before each study. The study conformed to the standards set by the Declaration of Helsinki and Good Clinical Practice guidelines.

A total of 36 EEG recordings were performed. Data are presented from 30 infants (0–3 months, n = 11; 4–6 months, n = 19) in the MOSSA analysis. Data are presented from 24 infants (0–3 months, n = 8; 4–6 months, n = 16) for emergence analysis. Details of the study profile and infant demographics are provided in *Figure 2* and *Table 1*, respectively.

## Anesthetic management

Each infant received anesthetic induction with sevoflurane ± nitrous oxide. Nitrous oxide was added at the discretion of the anesthesiologist (29/36 cases). Nitrous oxide was discontinued after placement of an endotracheal tube or laryngeal mask. Propofol was used to facilitate tracheal intubation in 19/36 cases (mean concentration ± standard deviation (SD): 15.8 ± 6.3 mg/kg; range 10–30 mg/kg). No infants were prescribed midazolam or other premedication on the day of surgery. Clinical characteristics for individual infants are given in *Supplementary file 1*.

MOSSA was comprised of sevoflurane administration with air and oxygen, titrated to clinical signs; end-tidal sevoflurane concentration was adjusted according to the anesthesiologist's impression of clinical need, not according to pre-set end-tidal sevoflurane concentration per IRB requirements for the pediatric risk category of 'minimal risk, no potential for benefit'. During MOSSA, the median end-tidal sevoflurane concentration for infants 0–3 months of age was 2.0% (95% CI: 0.8–2.6%, n = 11), and for infants 4–6 months of age was 2.6% (95% CI: 2.4–3.1, n = 19); *Table 1*.

## Data acquisition

All infants were in the supine position throughout the study. Each infant was studied once.

### EEG recording

An EEG cap was used to record EEG activity (WaveGuard EEG cap, Advanced NeuroTechnology, Enschede, Netherlands). 33 recording electrodes were positioned according to the modified international 10/20 electrode placement system at Fz, FPz, FP1, FP2, F3, F4, F7, F8, FC1, FC2, FC5, FC6, Cz, CPz, C3, C4, CP1, CP2, CP5, CP6, Pz, P3, P4, P7, P8, T7, T8, M1, M2, POz, Oz, O1, and O2 (*Figure 1*). Reference and ground electrodes were located at Fz and AFz, respectively. The impedance of the electrode-skin interface was kept to a minimum by massaging the skin with an EEG prepping gel (Nu-Prep gel, DO Weaver & Co., CO, USA), and conductive EEG gel was used to optimize contact with the electrodes (Onestep-Clear gel, H+H Medical Devices, Dülmen, Germany).

EEG activity from 0 to 500 Hz was recorded with an Xltek EEG recording system (EMU40EX, Natus Medical Inc., Ontario, Canada). Signals were digitized at a sampling rate of 1024 Hz (or 256 Hz in one case), and a resolution of 16 bit.

### Clinical data collection

Demographics and clinical information, including age, gender, surgical procedure, anesthetic management, were collected from the electronic medical records and from the in-house Anesthesia Information Management System (AIMS). End-tidal sevoflurane, oxygen, and nitrous oxide concentrations were downloaded from the anesthetic monitoring device (Dräger Apollo, Draeger Medical Inc., Telford, PA) to a recording computer in real-time using ixTrend software (ixellence, Wildau, Germany). Signals were recorded at a sampling rate of 1 data point per second. Gross body movement was recorded with a camcorder that was time-locked to the EEG recording (Xltek DSP270x, Natus Medical Inc.).

## EEG analysis

### Data preprocessing

Preprocessing was carried out with Natus NeuroWorks (Natus Medical Inc.) and in-built MATLAB code (MathWorks Inc., Natick, MA). Ear electrodes (M1 and M2) were excluded from the final analysis due to poor surface-to-skin contact for the majority of infants.

EEG signals were re-montaged to a nearest neighbor Laplacian reference using distances along the scalp surface to weight neighboring electrode contributions. We applied an anti-aliasing filter of 80 Hz and down-sampled the EEG data to 256 Hz.

EEG dynamics were analyzed at three distinct periods of (1) MOSSA, (2) awake (pre-anesthesia), and (3) emergence from general anesthesia. (1) *MOSSA analysis*: in the first part of this study, we were interested in characterizing EEG features during MOSSA. For each infant, a 10-min EEG segment during a period of maintenance of general anesthesia adequate for surgery and where end-tidal

sevoflurane concentration was maintained at a constant concentration ($\pm$0.1%) was identified. Within this time segment, a 5-min epoch was selected from 'artifact-free' EEG, where motion or electrocautery artifacts were not present in the EEG. Channels with noise or artifacts were excluded from the analysis by visual inspection. EEG data analyzed during MOSSA involved epochs with median time after propofol or nitrous oxide administration of 77 min (IQR 32–93 min). (2) *Awake state to MOSSA analysis*: in the second part of this study, we were interested in characterizing the relationship with EEG features during the awake state, and comparing them to MOSSA. For each infant, the EEG and video recording starting before any anesthetics were administered was identified. Each video was reviewed and the sleep state of the infant identified using behavioral and vocal markers. The awake state was defined as having one of three features (1) eyes open, (2) body movement, and/or (3) crying. For each corresponding EEG segment, we extracted an 11-s 'artifact-free' epoch by visual inspection. If there was no 'artifact-free' data available then we did not include that epoch in the analysis. (3) *Emergence from general anesthesia analysis*: in the third part of this study, we were interested in characterizing relationship with EEG features, end-tidal sevoflurane concentration and onset of body movement. For each infant, an EEG segment starting from 10 min before sevoflurane gas was turned off, and finishing when end-tidal sevoflurane concentration was 0% was identified. For each EEG segment, we extracted a 30-s 'artifact-free' epoch by visual inspection at 10 end-tidal sevoflurane concentration ranges: (1) 0 to <0.3%; (2) 0.3 to <0.6%; (3) 0.6 to <0.9%; (4) 0.9 to <1.2%; (5) 1.2 to <1.5%; (6) 1.5 to <1.8%; (7) 1.8 to <2.1%; (8) 2.1 to <2.4%; (9) 2.4 to <2.7%; and (10) 2.7 to <3.0%. A 15- or 20-s epoch was used in three occasions. If there was no 'artifact-free' data available in a specific concentration range then we did not include that epoch in the analysis.

## Time-frequency analysis

Spectral analysis of activity was performed with multitaper methods using the Chronux toolbox (http://chronux.org) (*Bokil et al., 2010*). Multitaper parameters were set using window lengths of T = 2 s with 1.9 s overlap; time-bandwidth product TW = 2, and number of tapers K = 3. The spectrum of frequencies over time within the 0–30 Hz range was plotted for individual electrodes in each infant.

Infants were divided into two groups according to postnatal age: (1) 0–3 months and (2) 4–6 months. First, group-averaged spectrograms were computed by taking the median power across subjects at each time and frequency at the electrode of interest for each postnatal age cohort (i.e., *Figure 4A,B*). Second, group-averaged spectra were computed by taking the mean power of individual spectrograms at each frequency across the entire epoch (5 min for MOSSA, and 30 s for emergence analysis), and then the median (IQR) power at each frequency was calculated for each postnatal age group (i.e., *Figure 4C*).

To identify the topographic distribution of specific frequency bands, we first took the mean power spectra of group-averaged spectrograms across the entire epoch (5 min for MOSSA and 30 s for emergence analysis) for each electrode. Then, we averaged the mean power spectra over each EEG frequency band of interest for both postnatal age groups. Scalp EEG frequency band power plots were performed using 3D interpolation of the electrode montage with the topoplot function in EEGLab (*Delorme and Makeig, 2004*).

## Coherence analysis

Coherence analysis was performed using custom-written MATLAB code (MathWorks Inc.; *Source code 1*). Coherence quantifies the degree of correlation between two signals at a given frequency. It is equivalent to a correlation coefficient indexed by frequency: a coherence of 1 indicates that two signals are perfectly correlated at that frequency, while a coherence of 0 indicates that the two signals are uncorrelated at the frequency. The coherence between channel i and j is given by

$$C_{ij}(f,t) = \left| \frac{S_{ij}(f,t)}{\sqrt{S_i(f,t)S_j(f,t)}} \right|,$$

where $S_{ij}$ is the cross-spectrum of i, jth electrode and $S_i$ is the spectrum of ith electrode at frequency *f* and time *t*. The spectrum is calculated by using multitaper methods with TW = 3, K = 5, T = 2 s, and a non-overlapping window. A coherogram graphically illustrates coherence for a range of frequencies plotted across time.

For *frontal coherence analysis*, we calculated and plotted coherence between electrodes F7 and F8, based on analyses using these electrodes in adult studies (*Akeju et al., 2014c*). To compute the

group-averaged frontal coherogram, we took the median coherence across infants at each time point and frequency. For *global coherence analysis*, we divided each segment into non-overlapping 2-s windows and computed the cross-spectral matrix. To remove noise artifact, the median over 10 windows of the real and imaginary parts of each entry in the cross-spectral matric was taken. Then, we performed an eigenvalue decomposition analysis of the cross-spectral matrix at each frequency. The cross-spectral matrix at each frequency can be factorized as

$$S(f) = U(f)\Lambda(f)U(f)^H,$$

where $U^H$ is the complex conjugate transpose of $U$ and a unitary matrix whose *i*th column is the eigenvector $u_i$ of $S$, and $\Lambda$ is the diagonal matrix whose diagonal elements are the corresponding eigenvalues, $\Lambda_{ii} = \lambda_i$. The global coherence is the ratio of the largest eigenvalue to the sum of eigenvalues (*Cimenser et al., 2011*):

$$C_{\text{global}}(f) = \frac{\lambda_{\max}(f)}{\sum_{i=1}^{N}\lambda_i(f)}.$$

When the largest eigenvalue is large compared with the remaining ones, the global coherence is close to 1. We computed the global coherence at each frequency using 5-min epochs for MOSSA. To get group-averaged global coherence plot, we took a median across infants. We refer to the eigenvector $u_{max}$ corresponding to the largest eigenvalue $\lambda_{max}$ at a given frequency as the principal mode of oscillation for that frequency and the coherence of electrode sites was obtained by the absolute value square of the eigenvector (*Purdon et al., 2013*). This means that the eigenvector describes a coherent spatial distribution. Spatial coherence at each frequency band was computed by taking an average across the frequency range at each electrode. Group-averaged scalp coherence distribution was computed by taking the median across infants and displayed using the topoplot function in EEGLab.

## Behavioral analysis

Behavioral data during (1) the awake state and (2) emergence from general anesthesia were analyzed post hoc. (1) *For awake analysis*: videos were reviewed frame-by-frame prior to anesthesia to identify the sleep state of each infant. The sleep state of the infant was determined using behavioral and vocal markers. The awake state was defined as from one of three features (a) eyes open, (b) body movement, and/or (c) crying. (2) *For emergence analysis*: videos were reviewed frame-by-frame to identify the time point (in seconds) where gross body movement first occurred. The corresponding end-tidal sevoflurane concentration was extracted from the ixTrend data recording. The percentage of infants who displayed gross body movement in each end-tidal sevoflurane concentration range was evaluated for each group.

## Statistical analysis

Data are shown as median (95% CI of median) unless otherwise stated. Statistical analysis performed using SPSS Statistics v.21 (IBM, Armonk, NY) and custom-written MATLAB code (MathWorks Inc.; *Source code 2*).

To assess statistical significance for the difference in power at each frequency, we computed the 95% CI by using a frequency domain-based bootstrapping algorithm (*Kirch and Politis, 2011*). We drew Fourier coefficients from normal Gaussian distribution with variance of its spectral power for each subject. From the Fourier coefficients, we computed replicates of spectral power for each subject and took the median value (i.e., power, or coherence, where relevant) across infants within each postnatal age group and computed group differences (using paired comparisons, where relevant), following this a new set was randomly selected in each postnatal age group and the analysis repeated. We repeated this 2000 times and calculated the 95% CI using the median difference at each frequency.

### Emergence analysis

To evaluate the power difference at each frequency between MOSSA and after the first movement, we first identified 'artifact-free' epochs for analysis. For MOSSA data, we extracted a 10- to 20-s 'artifact-free' epochs by visual inspection around 10 min before gas-off. For movement data, we

selected a 5- to 30-s 'artifact-free' epoch within 2 min after the onset of body movement. We computed the 95% CI using the bootstrap analysis of the difference between the median power spectra across infants in MOSSA data and movement data in each age group.

## Code

Custom-written MATLAB code (with simulated data) for computing global coherence (*Source code 1*), and multitaper spectra and bootstrap CIs (*Source code 2*) is given in the Supplemental Materials.

## Acknowledgements

We thank the Pre-operative and Operating Room staff at Boston Children's Hospital for their assistance during these studies, as well as the parents and infants who took part in the study.

## Additional information

### Competing interests

PLP: PLP has patents pending on brain monitoring during general anesthesia and sedation, and a patent licensing agreement with Masimo Corporation. Application Numbers: 20150080754, 20150011907, 20140323898, 20140323897, 20140316218, 20140316217, 20140187973, 20140180160, 20080306397. ENB: ENB has patents pending on brain monitoring during general anesthesia and sedation, and a patent licensing agreement with Masimo Corporation. Application Numbers: 20150080754, 20150011907, 20140323898, 20140323897, 20140316218, 20140316217, 20140187973, 20140180160. ENB is a Reviewing Editor for *eLife*. The other authors declare that no competing interests exist.

### Funding

| Funder | Grant reference | Author |
|---|---|---|
| Sara Page Mayo Endowment for Pediatric Pain Research and Treatment | | Laura Cornelissen, Charles B Berde |
| Boston Children's Hospital (BCH) | | Laura Cornelissen, Charles B Berde |
| National Institutes of Health (NIH) | R01-GM104948; DP2-OD006454 | Seong-Eun Kim, Patrick L Purdon, Emery N Brown |
| Massachusetts General Hospital (MGH) | | Patrick L Purdon, Emery N Brown |

The funders had no role in study design, data collection and interpretation, or the decision to submit the work for publication.

### Author contributions

LC, CBB, The work presented here was undertaken in collaboration between all authors. LC and CBB designed and performed the study. SK, PLP and ENB designed the computational analysis. SK and LC carried out the data analysis. All authors contributed to data analysis and data interpretation. LC, SK, CBB and ENB wrote the initial manuscript; all authors edited and revised the manuscript. All authors approved the final version of this manuscript, Conception and design, Acquisition of data, Analysis and interpretation of data, Drafting or revising the article; S-EK, PLP, ENB, The work presented here was undertaken in collaboration between all authors. LC and CBB designed and performed the study. SK, PLP and ENB designed the computational analysis. SK and LC carried out the data analysis. All authors contributed to data analysis and data interpretation. LC, SK, CBB and ENB wrote the initial manuscript; all authors edited and revised the manuscript. All authors approved the final version of this manuscript, Analysis and interpretation of data, Drafting or revising the article

### Author ORCIDs

Laura Cornelissen, http://orcid.org/0000-0001-8579-0870

## Ethics

Human subjects: Boston Children's Hospital Institutional Review Board (IRB) approved the study (IRB Protocol Number: IRB-P00003544), and informed written consent was obtained from parents/legal guardians before each study. The study conformed to the standards set by the Declaration of Helsinki and Good Clinical Practice guidelines.

## Additional files

### Supplementary files

• Supplementary file 1. Characteristics of individual infants. Subjects listed according to postnatal age. Data given for all infants included in the MOSSA analysis. M, months; MOSSA, Maintenance Of a Surgical State of Anesthesia; PNA, Postnatal Age; Wgt., weight. A–All infants were administered glycopyrrolate-neostigmine to reverse the neuromuscular blockade towards the end of surgery (except infants 1, 3, and 15). B–Infant was additionally administered clonidine for hypertension.

• Source code 1. Custom-written MATLAB code (with simulated data) for computing global coherence.

• Source code 2. Custom-written MATLAB code (with simulated data) for computing multitaper spectra and bootstrap CIs.

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
