## [Decision Letter]

Thank you for sending your work entitled “Age-dependent electroencephalogram (EEG) patterns during sevoflurane general anesthesia in infants” for consideration at *eLife*. Your article has been favorably evaluated by Eve Marder (Senior editor) and three reviewers, one of whom, Jody Culham, is a member of our Board of Reviewing Editors.

The Reviewing editor and the other reviewers discussed their comments before we reached this decision, and the Reviewing editor has assembled the following comments to help you prepare a revised submission.

Main revisions needed:

All three reviewers found the study interesting and important; however, there are several issues that must be addressed in a revision. The key changes required are summarized in the numbered points below. In addition specific comments from Reviewer #2 are also appended to provide additional detail.

1) One expert reviewer brought up a number of important concerns about the brain states of infants during different stages. Specifically, the revision should address the validity of using movement as a proxy for emergence, the possibility of sleep during emergence, and the lack of data from the pre-anesthesia states. See Reviewer #2's detailed points below.

2) The possibility that anesthetic variances could explain the data should be explicitly discussed. This includes explicit statistics to determine whether the weights and durations of anesthesia in the two age groups differed significantly. See Reviewer #2's detailed points below.

3) More statistical comparisons of the putative differences are needed. See Reviewer #2's detailed points below.

4) One reviewer suggests one area that can be addressed in the Discussion is the role, if any, of the developmental change in the grey/white matter ratios that have been documented in infant MRI studies. This may provide a clinically measurable structural basis for the neurophysiological changes observed with sevoflurane anesthesia in this report.

5) The authors should reconsider the framing of the conclusions of the paper, although there were some conflicting views about this.

One main suggestion, which came up during the initial manuscript consultation amongst *eLife* editors was the need to make it clearer what the relevance/implications of the results are not just for clinical practice but for understanding brain development. *eLife* is a biological sciences journal so we are looking for papers that advance our understanding of the underlying biology. There is some interesting discussion about possible reasons for the differential effects of anesthesia in younger infants, older infants and adults. However, this is somewhat underplayed in the manuscript as it stands. For example, the Abstract says, “these differences in EEG dynamics are likely due to brain development”, but doesn't unpack why these are interesting.

That said, given some of the concerns raised by reviewers, we do not want to push you to draw stronger conclusions than the data warrant, particularly if some of the differences may relate to potential confounds such as anesthesia levels. One reviewer also wonders if the conclusions in the final sentence of the Abstract are too strong. See Reviewer #2's detailed points below (regarding the Abstract).

*Detailed comments from Reviewer #2*:

This is a really interesting study that is challenging to perform and timely in terms of the results generated and their relevance. I am generally enthusiastic about this paper, but more work is required to be convinced by the authors' main interpretation and explanation of their results. Namely, that the developing brain is the most likely explanation for their convincing and well analysed EEG results showing clear differences during surgical anaesthesia between 0-3 and 4-6 month old babies. I describe below in more detail this issue alongside minor other concerns.

1) Discussion, first paragraph. Please expand on the discussion of the flaws in using only this behavioural measure (see more below). Is it possible to relate any other measure (respiration/heart rate change) to these crude movement behavioural measures so that we can be more sure that the movement as measured relates to something that additional suggests true emergence/awake-like state (see below for more on this and complexity of sleep stages).

2) In the second paragraph of the subsection headed “Studies of EEG dynamics of infant brain under general anesthesia”. Is this finding better explained by <3 month-old babies being more likely to enter a sleep stage during emergence? The authors admit this is possible later in the manuscript. This confound needs more discussion and to be offered as a viable additional explanation perhaps.

3) In the subsection headed “Studies of EEG dynamics in adults under general anesthesia”. The lack of pre-anaesthetised baseline data is key. While understandably a complex environment, presumably the electrodes were already in place and so I am unclear why these data were not taken prior to induction of anaesthesia? How can you relate movement to ‘awake’ EEG if you don't know what genuine awake EEG looks like in these different age groups? These data are needed surely to make sense of anaesthetised states—possibly it's the relative EEG changes from fully awake to anaesthetised in each baby that are as relevant as the between ages changes during ‘assumed anaesthetic dose/state’?

4) In the subsection headed “Anesthetic Management”. Where the methods are described we begin to unravel some of the major confounds here. These need to be brought earlier into the Introduction and Results and explanation of findings. They are buried here. Please include the anaesthetic data in Table 1 and provide statistics. Clearly there are many differences here—understandable, as routine clinical care was performed and this is an opportunistic uncontrolled study, but as such the authors must alert the readership to the likelihood that anaesthetic variances might largely explain their data. As written, this information is not highlighted or discussed and the reader is very strongly led to believe this is all due to the very interesting developmental changes. I don't doubt that there is a profound developmental influence, but we need to be convinced these other factors are not relevant. Firstly, propofol was used for some babies but not others. We do not know whether this was evenly distributed between the two age groups or not. Please detail. There are differences in duration and amount of anaesthetic used between age groups, but again we're not told whether this was significantly different for the two age groups. These factors alone will confound the EEG signals; therefore, please confirm whether they are significantly different or not—hopefully not and then we can be more convinced by their developmental explanation.

5) Figure 5. While frontal-occipital power is not different there does appear to be more power in the slow waves in the temporal regions; is this significant?

6) Figure 7. Again, temporal lobes ‘look’ different but we're not given any statistical comparisons. I wonder whether the likely developing insula cortex within the temporal lobes might be relevant here?

7) Figure 10. Can we segregate body movements into those with eyes open/closed or whether concomitant with being obviously awake, etc? No information is given about this crude measure, its variance and therefore whether there were systematic differences in the age groups. For example, can we eliminate the question regarding whether babies were still predominately asleep in 0-3 month group even with body movements compared to the 4-6 month olds? Also, what does the body movement do to the quality of EEG recordings and were any body movements of the head? More detail is needed considering this was the single behavioural anchor used to determine emergence and to time-lock the EEG analysis to.

8) Table 1. We need statistics here. There is a large weight difference and differences in length of anaesthesia, which means the brains of younger babies were loaded with more anaesthetic and this will impact emergence. Was this a significant difference? More details needed about the reasons for surgery—any brain/spinal cord related? Please also include from Methods into this table the dose/duration details regarding all anaesthetic agents used with statistics.

9) Abstract: I don't think you can make such a strong conclusion at this stage and based upon your conclusion. You might live to regret stating such a limiting statement regarding the current likelihood that EEG indices could be used. Arguably they might provide excellent measures once you know (and can factor in) the neurophysiological state of brain from which they derive. Please alter the final sentence.

---

## [Author Response]

*1) One expert reviewer brought up a number of important concerns about the brain states of infants during different stages. Specifically, the revision should address the validity of using movement as a proxy for emergence, the possibility of sleep during emergence, and the lack of data from the pre-anesthesia states. See Reviewer #2's detailed points below*.

Please see our response to Reviewer #2 (Comment # 1, 2, 3 and 7).

*2) The possibility that anesthetic variances could explain the data should be explicitly discussed. This includes explicit statistics to determine whether the weights and durations of anesthesia in the two age groups differed significantly. See Reviewer #2's detailed points below*.

Please see our response to Reviewer #2 (Comment # 4 and 8).

*3) More statistical comparisons of the putative differences are needed. See Reviewer #2's detailed points below*.

Please see our response to Reviewer #2 (Comment # 5 and 6).

*4) One reviewer suggests one area that can be addressed in the Discussion is the role, if any, of the developmental change in the grey/white matter ratios that have been documented in infant MRI studies. This may provide a clinically measurable structural basis for the neurophysiological changes observed with sevoflurane anesthesia in this report*.

Structural MRI images from infants under 6 months show relatively high water content of both grey and white matter at this age. Postmortem studies show that significant myelination occurs postnatally and proceeds in a posterior-to-anterior fashion. Myelination begins in the pons and cerebellar peduncles, and by 3 months has extended to the optic radiation and splenium of the corpus callosum. At around 8-12 months of age, the white matter associated with the frontal parietal and occipital lobes becomes apparent. The emergence of frontal alpha activity at around 3 to 4 months postnatal ages likely parallels the regional changes in cortico-cortical and thalamocortical maturation. Please see the subsection “Brain development as a source of changes in EEG dynamics with age” in the Discussion.

*5) The authors should reconsider the framing of the conclusions of the paper, although there were some conflicting views about this*.

*One main suggestion, which came up during the initial manuscript consultation amongst* eLife *editors was the need to make it clearer what the relevance/implications of the results are not just for clinical practice but for understanding brain development.* eLife *is a biological sciences journal so we are looking for papers that advance our understanding of the underlying biology. There is some interesting discussion about possible reasons for the differential effects of anesthesia in younger infants, older infants and adults. However, this is somewhat underplayed in the manuscript as it stands. For example, the Abstract says, “these differences in EEG dynamics are likely due to brain development”, but doesn't unpack why these are interesting*.

Please see our response to Reviewer #2 (Comment # 9).

*That said, given some of the concerns raised by reviewers, we do not want to push you to draw stronger conclusions than the data warrant, particularly if some of the differences may relate to potential confounds such as anesthesia levels. One reviewer also wonders if the conclusions in the final sentence of the Abstract are too strong. See Reviewer #2's detailed points below (regarding the Abstract)*.

Detailed comments from Reviewer #2:

*This is a really interesting study that is challenging to perform and timely in terms of the results generated and their relevance. I am generally enthusiastic about this paper, but more work is required to be convinced by the authors' main interpretation and explanation of their results. Namely, that the developing brain is the most likely explanation for their convincing and well analysed EEG results showing clear differences during surgical anaesthesia between 0-3 and 4-6 month old babies. I describe below in more detail this issue alongside minor other concerns*.

*1) Discussion, first paragraph. Please expand on the discussion of the flaws in using only this behavioural measure (see more below). Is it possible to relate any other measure (respiration/heart rate change) to these crude movement behavioural measures so that we can be more sure that the movement as measured relates to something that additional suggests true emergence/awake-like state (see below for more on this and complexity of sleep stages)*.

We agree with the reviewer that there are uncertainties and limitations with a variety of behavioral and physiologic measures of recovery from the anesthetic state. In studies of adult volunteers, response to verbal command is often used; this is not feasible for infants. The advantage of using gross movement as a measure of emergence is that it is well-defined in infants and children (e.g. Bould et al., 2011 Pediatric Anesthesia; McCann et al., 2002; Pediatric Anesthesia; Davidson et al., 2001 Anesthesia & Analgesia). Onset of movement relates to the anesthetic state, namely the ability of general anesthetics to suppress movement. Respiration, heart rate and blood pressure also reflect effects of anesthesia but in a more complex and derivative manner. Each of these parameters can be influenced by anesthetic state, but also can be influenced by a number of factors summarized in Table 2 below. As per our IRB protocol, to make this a minimal risk study of infants, we had to leave all aspects of drug administration to the anesthesiologist’s discretion.

Author response table 1.Factors that challenge use of non-behavioral measures during emergence**DOI:**
http://dx.doi.org/10.7554/eLife.06513.023**Influencing factors****Reason****Relation to our study**Amount of painful stimulation from surgery affecting respiration & heart rate changeVariability in degree of suppression from nociceptive afferent drive by:• Local or regional anesthesia (e.g. field blocks for hernia repairs, or penile blocks for hypospadias repair)• Opioid dosing (dose and time course)[Supplementary-material SD1-data] provides details on the individual infant clinical data.Control of respiratory function• Variability in tidal volume and rate delivered via mechanical ventilation during anesthesia according to individual anesthesiologist’s clinical decisions. Resulting pCO2 influences ventilatory drive during emergence.• Variability in anesthesiologist’s method to convert from controlled to assisted or spontaneous ventilationInformation unavailable.Pharmacologic effects on heart rateVariability in use of neuromuscular blockade and need for reversal of neuromuscular blockade (case-dependent):• Reversal of NM blockade requires cholinesterase inhibitors (e.g. neostigmine), which produce bradycardia, and muscarinic antagonists (i.e. atropine or glycopyrrolate), which produce tachycardia. Heart rate during emergence can reflect individual differences in responses to these, along with other contributors to sympathetic and parasympathetic heart rate modulation.• Variability in timing of NM block reversal drug administration at the end of surgery, and effects on heart rate[Supplementary-material SD1-data] provides details on the individual infant clinical data.

We collected respiration rate, heart rate and non-invasive blood pressure data during emergence. Mean respiration rate and heart rate over a 20-minute period of emergence are shown for all infants in Figure 13. The onset of body movement is marked as time 0.

Author response image 1.Onset of body movement is not associated with changes in respiration rate.(A) Mean respiration rate (breaths per min) and (B) heart rate were calculated for all infants over a 20 minute period: 10-min prior to first visible body movement, and 10min after. Time 0 (grey dashed line) indicates where first body movement was observed. Data represented as mean (+/- SD).**DOI:**
http://dx.doi.org/10.7554/eLife.06513.024

Respiration rate data was sampled at every 1 min using a clinical movement transducer. The low temporal resolution meant that subtle changes in respiration rate at the time of first body movement were not detectable adequate second-by-second accuracy (Figure 13). Based on the available data, we show there were no changes in respiration rate at 1 min before body movement compared to 1 minute after body movement (p=0.56, paired Student’s t-test; Table 3).

Author response table 2.Change in respiration rate and heart rate at the time of first body movement**DOI:**
http://dx.doi.org/10.7554/eLife.06513.025Baseline(1min prior to first body movement)Post-movement(1min after first body movement)**p-value**^A^**Respiration rate (breaths per min)**All infants0-3M4-6M29.4 (13.4)36.3 (14.7)23.5 (11.5)30.8 (12.1)39.1 (13.3)25.7 (8.1)0.590.440.90**Heart rate (bpm)**All infants0-3M4-6M145 (23)156.4 (21.7)139.6 (22.1)150 (25.3)165 (22)143.9 (24.4)**0.0018 (**)****0.042 (*)****0.023 (*)**

Data in parenthesis represent standard deviation. Bpm, beats per min; M, months old; min, minute. A – Paired Student’s t-test.

Asterisks indicate level of significance where: *, p<0.05; and **, p<0.01.

Onset of body movement was associated with increased heart rate in all infants. We show that mean HR over 1 min period immediately prior to first visible body movement was 145 bpm (SD: 23) and significantly increased during first body movement [Mean HR: 150 bpm (SD: 25.3); p=0.0018, paired Student’s t-test; Table 3]. Non-invasive blood pressure was taken every 5mins using an automated pressure cuff. The low sampling rate meant that blood pressure measurements were taken before and after first body movement within time fame of +/- 5 mins. Therefore, not all subjects had a blood pressure sample taken at both 1 min prior to- and at 1 min after first visible body movement; this parameter could not be included in the analysis.

We have edited the Discussion section “Study Limitations and Constraints” to include the following: “In place of formal behavioral assessments of consciousness, which would not be feasible in children […] also reflect effects of anesthesia but in a more complex manner being dependent on drug administration and clinical state of the patient. ”

*2) In the second paragraph of the subsection headed “Studies of EEG dynamics of infant brain under general anesthesia”. Is this finding better explained by <3 month-old babies being more likely to enter a sleep stage during emergence? The authors admit this is possible later in the manuscript. This confound needs more discussion and to be offered as a viable additional explanation perhaps*.

We used onset of behavioral movement as a surrogate marker for onset of emergence, and this has been used with other behavioral measures such as crying (or attempting to cry), gagging on an endotracheal tube, eye opening and looking around to describe awakening in neonates and infants (Bould et al., 2011 Pediatric Anesthesia). All infants included in the emergence analysis required extubation. The standard anesthetic practice to extubate uses one of two ways: (1) “Awake with movement”, meaning with body movement, coughing, and return of some reflex activity, or (2) “Deep”, meaning with spontaneous breathing, but in a deeper plane of anesthesia, specifically without coughing. For >90% of infant anesthetics at Boston Children’s Hospital, awake extubation is preferred, although anesthesiologists differ in the degree of responsiveness required prior to extubation. It is plausible that some infants may remain sedated or unconscious despite recovering movement immediately following surgery. Nevertheless, our comparisons of the EEG properties between arousal states indicate striking similarities between the awake state and emergence in infants 0 to 3 months (as well as between the awake state and MOSSA). The implications of these findings demonstrate the challenges faced when using surface EEG measures to evaluate anesthetic depth in very young infants.

*3) In the subsection headed “Studies of EEG dynamics in adults under general anesthesia”. The lack of pre-anaesthetised baseline data is key. While understandably a complex environment, presumably the electrodes were already in place and so I am unclear why these data were not taken prior to induction of anaesthesia? How can you relate movement to ‘awake’ EEG if you don't know what genuine awake EEG looks like in these different age groups? These data are needed surely to make sense of anaesthetised states—possibly it's the relative EEG changes from fully awake to anaesthetised in each baby that are as relevant as the between ages changes during ‘assumed anaesthetic dose/state’*?

Collecting “awake pre-anesthesia” EEG data from infants is technically and logistically very challenging. Ideal recording conditions require the infant to be relaxed i.e. well fed, lying still, and in a cot. These conditions help avoid EEG signal contamination from muscle and movement, and external electrical noise. Most infants in the preoperative holding area were hungry (having abstained from feeding at least 4-6 hours prior to surgery per anesthesiologist requirement for an “empty stomach” to reduce the risk of tracheal aspiration of gastric contents). They were easily agitated and consequently episodically physically active during the Awake EEG recordings. Those infants who required comforting were often held in a parent’s arms; this also generated external electrical noise.

Following the reviewer’s suggestion, we now provide awake- state data analysis in the revised report. For the reasons listed above, we made extensive effort to identify good artifact-free preoperative epochs, and used 11-second samples of artifact-free EEG data for the awake-state analysis. Awake-state data were evaluated in each group to identify age-dependent differences, and compared to 11-second samples of MOSSA data to identify sevoflurane general anesthesia dependent differences.

Our findings show that slow and delta activity are the prominent frequencies during the awake state at all ages. There were no discernable changes in power between the awake state and MOSSA in infants aged 0 to 3 months. However, increased theta (>6Hz) and alpha were shown during the MOSSA state compared to the awake-state in infants 4 to 6 months of age. These findings suggest that the emergence of alpha activity is age-dependent, and a marker of anesthetic induced unconsciousness in infants 4 to 6 months.

The Methods section is now edited to reflect the inclusion of awake- state data. The results include an additional section detailing the awake-state data analysis with comparisons to the “MOSSA” data (Figure 10 and Figure 10—figure supplement 1).

*4) In the subsection headed “Anesthetic Management”. Where the methods are described we begin to unravel some of the major confounds here. These need to be brought earlier into the Introduction and Results and explanation of findings. They are buried here. Please include the anaesthetic data in*
Table 1
*and provide statistics. Clearly there are many differences here—understandable, as routine clinical care was performed and this is an opportunistic uncontrolled study, but as such the authors must alert the readership to the likelihood that anaesthetic variances might largely explain their data. As written, this information is not highlighted or discussed and the reader is very strongly led to believe this is all due to the very interesting developmental changes. I don't doubt that there is a profound developmental influence, but we need to be convinced these other factors are not relevant. Firstly, propofol was used for some babies but not others. We do not know whether this was evenly distributed between the two age groups or not. Please detail. There are differences in duration and amount of anaesthetic used between age groups, but again we're not told whether this was significantly different for the two age groups. These factors alone will confound the EEG signals; therefore, please confirm whether they are significantly different or not—hopefully not and then we can be more convinced by their developmental explanation*.

We have edited Table 1 to include the requested information. This includes: (1) detail from the methods i.e. end-tidal sevoflurane concentration during MOSSA epochs; (2) general anesthetic management i.e. number of subjects given propofol, duration of anesthetic exposure; and (3) statistical comparisons. Table 1 also cross-references to a supplementary table detailing individual infants’ demographics, clinical characteristics and anesthetic management.

The findings related to this study are possibly confounded by differences in anesthetic depth among individuals during MOSSA. For example, the addition of propofol to the anesthetic or increase in end-tidal sevoflurane concentration can potentially alter the depth of anesthesia as well as the characteristics of the EEG spectra.

End-tidal sevoflurane concentration requirements during MOSSA were significantly lower in the 0-3 month infants compared to the 4-6 month infants (p=0.002, Mann-Whitney U-test; Table 1). We performed a secondary analysis to control for potential dose effects by comparing EEG spectra during MOSSA at identical end- tidal sevoflurane concentrations of 1.8% across infants. At this lighter concentration (compared to MOSSA), the patterns of EEG activity observed were also seen during MOSSA suggesting the observed differences between the two groups were not likely due to a dose effect. We have edited the Results section (see EEG features during MOSSA) as follows:

“End-tidal sevoflurane requirements during MOSSA were significantly lower in infants 0 to 3 months compared to infants 4 to 6 months of age (Table 1). Therefore, we performed a secondary analysis to control for potential dose effects by comparing EEG spectra during MOSSA at identical end-tidal sevoflurane concentrations of 1.8% across infants. We chose 1.8% end-tidal sevoflurane because all of the infants received this concentration for at least 30s during surgery, and did not exhibit body movement or reflex activity at this plane of anesthesia suggesting a behavioral state comparable to MOSSA. At 1.8% end-tidal sevoflurane, frontal group-median spectrograms at F7 show slow and delta power was dominant over other frequencies in both age groups (Figure 4—figure supplement 1). Higher frequency oscillations in the theta and alpha range were more prominent in infants 4 to 6 months of age (Figure 4—figure supplement 1). The patterns of EEG activity observed during MOSSA were also seen during lighter planes of MOSSA at 1.8% end-tidal sevoflurane suggesting that the observed differences between the two groups are not likely due to a dose effect.”

We found no significant differences in the distribution of propofol administration between age groups (0-3M, 27.3%, 4- 6M, 42.1%; Fisher’s exact test, p=0.08; Table 1). There were no differences in the duration of anesthetic exposure between age groups (p=0.18, Mann-Whitney U test). Collectively, these findings indicate that although a lower dose of anesthetic was required to maintain a surgical state of anesthesia in the younger infants, the differences seen between the groups are less likely to reflect deeper levels of anesthesia and more likely to be age-dependent.

*5)*
Figure 5*. While frontal-occipital power is not different there does appear to be more power in the slow waves in the temporal regions; is this significant*?

We evaluated spectral power at the temporal regions and along the midline areas. Power in the slow frequencies was significantly greater in the left (T7) temporal region when compared to key midline electrodes located in the frontal (FPz), central (Cz) and occipital (Oz) regions in all infants (Figure 14). These findings that midline activity is compromised of significantly lower power compared to temporal regions are likely due to heterogeneity of skull thickness or bone conductance, and physical changes in the brain.

Author response image 2Midline EEG spectral power is lower compared to temporal regions during MOSSA in all infants.Temporal (T7) electrode comparisons are shown for (A-E) vs. frontal midline (FPz); (F-J ), vertex (Cz); (K-P), occipital midline (Oz); and (Q- U) lateral temporal area (T8). For each electrode pair, a schematic of electrode locations ( A, F, K, Q) are shown with the respective analysis of power spectra in infants aged 0-3 months ( B, G, L, R), and 4-6 months (C, H, M, S), and 95% CI, paired bootstrap analysis for infants 0-3 months (**D, I, O, T**), and 4-6 months of age (**E, J, P, U**).**DOI:**
http://dx.doi.org/10.7554/eLife.06513.026

In neonates and infants, the skull is thinner and more vascularized than in adults. The cranial bones are un-fused at birth, separated by fontanels and sutures for several postnatal months to accommodate the rapid growth of the brain. The fontanels are located at the midline junctions of bregma and lamda, and have a thinner zone in the skull; where the skull is thinner, it is more conductive. It is plausible that a high signal to noise ratio is obtained over these regions and contributes to these spatial differences in power.

*6)*
Figure 7*. Again, temporal lobes ‘look’ different but we're not given any statistical comparisons. I wonder whether the likely developing insula cortex within the temporal lobes might be relevant here*?

Please see our response to Point 5 above. We speculate that the predominance of lower frequency activity seen reflects less cortico-cortical generator activity, and more thalamocortical generator activity (Michelle de Haan, Infant EEG and Event-Related Potentials, Psychology Press, 2013). Future studies using complementary techniques are needed for accurate source localization (i.e. with high density EEG electrode arrays) and evaluation of the origin of deeper cortical and subcortical signals (i.e. fMRI, PET).

*7)*
Figure 10*. Can we segregate body movements into those with eyes open/closed or whether concomitant with being obviously awake, etc? No information is given about this crude measure, its variance and therefore whether there were systematic differences in the age groups. For example, can we eliminate the question regarding whether babies were still predominately asleep in 0-3 month group even with body movements compared to the 4-6 month olds? Also, what does the body movement do to the quality of EEG recordings and were any body movements of the head? More detail is needed considering this was the single behavioural anchor used to determine emergence and to time-lock the EEG analysis to*.

Please see our response to Point 2 above. The EEG power spectra in infants 0-3 months during MOSSA is comparable to that of emergence, as well as the awake state. The findings of our study demonstrate how at present, features of the EEG make it challenging to monitor anesthetic depth in infants aged 0-3 months of age, compared to older infants.

The reviewer is rightly concerned about the influence of body movement on the quality of EEG recordings. We made extensive effort to identify good artifact-free emergence epochs, and used 10-30-second samples of artifact-free EEG data for the Emergence analysis.

We anticipate performing future studies examining the time course of events during emergence using detailed analysis of cortical activity with behavioral (evoked and purposeful), and physiologic measures.

*8)*
Table 1*. We need statistics here. There is a large weight difference and differences in length of anaesthesia, which means the brains of younger babies were loaded with more anaesthetic and this will impact emergence. Was this a significant difference? More details needed about the reasons for surgery—any brain/spinal cord related? Please also include from Methods into this table the dose/duration details regarding all anaesthetic agents used with statistics*.

The patterns of uptake of inhalation anesthetics depend on time course of inhaled concentrations, alveolar ventilation, cardiac output and blood to gas and gas to oil partitioning. In steady-state, for an insoluble anesthetic such as sevoflurane, brain partial pressure is close to inspired partial pressure, with little dependence on body weight or brain weight. The anesthetic concentration was titrated by the anesthesiologist based on clinical considerations that we did not control. Nevertheless, the difference in mean end -tidal sevoflurane concentration between infants 0 to 3 months, and 4 to 6 months was small (Table 1). Previous studies of MAC for sevoflurane in infants showed a very slight age-dependence, with newborns and very young infants tending towards slightly lower MAC values compared to infants around 4 to 6 months of age. We have provided statistical information on the demographics and duration of anesthesia in Table 1 of the revised article. We show there were no differences in the duration of anesthetic exposure between the age groups (p=0.18, Mann-Whitney U test).

*9) Abstract: I don't think you can make such a strong conclusion at this stage and based upon your conclusion. You might live to regret stating such a limiting statement regarding the current likelihood that EEG indices could be used. Arguably they might provide excellent measures once you know (and can factor in) the neurophysiological state of brain from which they derive. Please alter the final sentence*.

The last sentence of the Abstract now reads: “We demonstrate the need to apply age-adjusted analytic approaches in order to develop neurophysiologic-based strategies for pediatric anesthetic state monitoring.”